# UniCalli: A Unified Diffusion Framework for Column-Level Generation and Recognition of Chinese Calligraphy

**Tianshuo Xu[1], Kai Wang[2], Zhifei Chen[1], Leyi Wu[1], Tianshui Wen[3], Fei Chao[3], Ying-Cong Chen[1,4,*]**
[1]HKUST(GZ), [2]China University of Geoscience Beijing, [3]Xiamen University, [4]HKUST
`txu647@connect.hkust-gz.edu.cn`; `yingcongchen@ust.hk`; (*Corresponding Author)

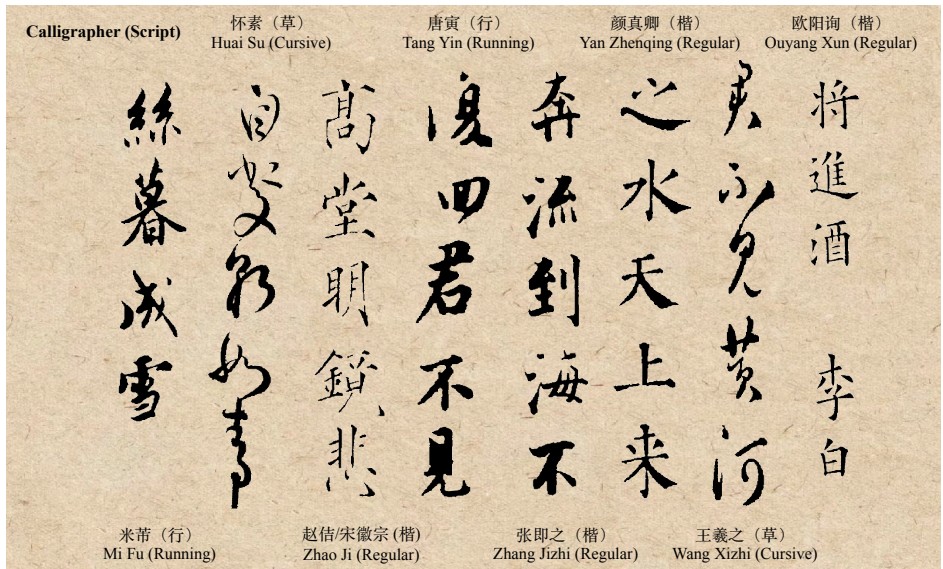

"Bring in the Wine" - Li Bai
Do you not see the Yellow River's waters pouring from the heavens?
They surge into the sea, never to return.
Do you not see in the bright mirrors of high halls a grief for white hair?
At dawn, it is like black silk; by dusk, it has turned to snow.

将进酒 李白
君不见，黄河之水天上来，奔流到海不复回
君不见，高堂明镜悲白发，朝如青丝暮成雪

Figure 1: Generated by the **UniCalli** model, this image displays calligraphy from Li Bai's poem "Bring in the Wine". Each column showcases a different master's style to demonstrate the model's versatility. Notably, especially in the Cursive script, the model generates contextually appropriate **connecting strokes** and **character sizes** based on adjacent characters. An English translation and the original Chinese text are provided in the lower corners. **The complete works by each calligrapher are available in the Appendix** I. (The calligraphic background has been manually edited for presentation.)

## Abstract

Computational replication of Chinese calligraphy remains challenging. Existing methods falter, either creating high-quality isolated characters while ignoring column-level aesthetics like ligatures and spacing, or attempting column synthesis at the expense of calligraphic correctness. We introduce **UniCalli**, a unified diffusion framework for column-level recognition and generation. Training both tasks jointly is deliberate: recognition constrains the generator to preserve character structure, while generation provides style and layout priors. This synergy fosters concept-level abstractions that improve both tasks, especially in limited-data regimes. We curated a dataset of over 8,000 digitized pieces, with 4,000 densely annotated. UniCalli employs asymmetric noising and a rasterized box map for spatial priors, trained on a mix of synthetic, labeled, and unlabeled data. The model achieves state-of-the-art generative quality with superior ligature continuity and layout fidelity, alongside stronger recognition. The framework successfully extends to other ancient scripts: Oracle bone scripts and Egyptian hieroglyphs.

# 1 INTRODUCTION

Chinese calligraphy is a central vehicle of Chinese culture and a world heritage art form, practiced and studied by millions (Yangbo, 2022; Nihon Shuji, 2025). While recent deep learning has produced recognition (Luo et al., 2025; Zhou et al., 2025) and generative (Liao et al., 2023; 2024; Wu et al., 2020) systems, progress is hindered by scarce data and a long-tail distribution of styles. Consequently, existing generative methods are limited. On one hand, isolated-character synthesis and font transfer techniques (Zhang et al., 2024; Yang et al., 2024; Xie et al., 2021) can produce high-quality individual characters but ignore the holistic aesthetics of a finished work: **column-level composition, spatial rhythm, and the crucial inter-character ligatures** that convey artistic intent. On the other hand, general image generation models (Labs et al., 2025; Esser et al., 2024) and VLM-based systems (Bai et al., 2025; OpenAI, 2024; Gong et al., 2025) that attempt column-level synthesis often **fail on correctness, rendering characters with improper forms or styles**. This leaves a critical gap for generating complete calligraphic works that are both structurally sound and artistically coherent.

To address these challenges, we contribute both a dataset and a model. **Dataset.** We curate a corpus of more than 8,000 digitized works spanning 93 classical calligraphers (e.g., Wang Xizhi, Mi Fu, Ouyang Xun). Over 4,000 works—covering hundreds of thousands of characters—are annotated with script type (regular/kai, running/xing, cursive/cao), per-character bounding boxes, and modern-character transcriptions. **Method.** We introduce **UniCalli**, a unified diffusion-based framework that learns jointly from synthetic, labeled, and unlabeled data, improving robustness in long-tail and limited-label regimes.

Unlike pipelines that separate recognition and generation, UniCalli integrates them in a single model with shared representations. This coupling is intentional: the recognition objective pressures the generator to preserve character identity and legibility, while the generative objective supplies strong style/layout priors and rich augmentations that make recognition more reliable across writers and scripts. The joint training encourages the model to form transferable, concept-level abstractions of characters (radicals, strokes, structures) that benefit both tasks and reduce label dependence. Conditioned on text, calligrapher identity, and script, UniCalli composes vertical, column-wise layouts with inter-character ligatures and deliberate control over character scale and spacing, producing complete-work outputs rather than isolated glyphs while maintaining stylistic consistency and character accuracy.

Architecturally, UniCalli builds on a Multimodal Diffusion Transformer (MMDiT) backbone (Blattmann et al., 2023), departing from causal autoregressive rollouts. During synthesis, the diffusion transformer attends bidirectionally over the full canvas at each step, enabling globally consistent layout decisions that mirror how calligraphers plan a page before committing strokes. To unify recognition and generation, we apply asymmetric noising to two coupled latents—a clean "standard-font" rendering of the target text and a calligraphy image. Noising the calligraphy branch while keeping the standard-font branch clean yields **generation**; reversing this configuration yields **recognition**. To strengthen spatial reasoning (character extents, column alignment, inter-character spacing), we augment the calligraphy input with a rasterized bounding-box map and jointly denoise the pair under a shared schedule, helping the model internalize position and scale priors that improve ligature formation and column rhythm.

Empirically, UniCalli performs strongly on both tasks. On recognition benchmarks, the unified model attains accuracy comparable to task-specialized recognizers. For generations, quantitative metrics and human evaluations indicate state-of-the-art results in glyph correctness and stylistic fidelity. Beyond Chinese calligraphy, we further validate the framework on Oracle bone inscriptions and Egyptian hieroglyphs, demonstrating adaptability across writing systems and highlighting its potential for the digitization and study of ancient scripts.

Our main contributions are threefold:

- **A Large-Scale Annotated Calligraphy Dataset**: We present a new, large-scale corpus of over 8,000 classical Chinese calligraphy works. More than 4,000 of these are annotated with script type, per-character bounding boxes, and modern transcriptions, providing a valuable resource to spur research in column-level analysis and generation.

- **UniCalli, a Unified Recognition and Generation Framework**: We propose a novel diffusion transformer model that, for the first time, unifies column-level calligraphy generation and recognition. Its bidirectional attention mechanism enables globally coherent composition, moving beyond isolated characters to produce complete, stylistically consistent works.
- **Demonstrated State-of-the-Art Performance and Generalizability**: We show that UniCalli achieves state-of-the-art results in generative fidelity and competitive performance in recognition. Furthermore, we validate its adaptability on other complex and ancient writing systems, including Oracle bone script and Egyptian hieroglyphs, demonstrating the broad potential of our approach.

## 2 RELATED WORK

### 2.1 DIFFUSION MODELS

Diffusion models (Ho et al., 2020; Song et al., 2020; Lipman et al., 2022) have recently emerged as a powerful class of generative models, achieving state-of-the-art results in high-fidelity image synthesis. Their core mechanism consists of two processes: a fixed forward process that gradually adds Gaussian noise to an input image until it becomes pure noise, and a learned reverse process. In the reverse process, a neural network, typically a U-Net (Rombach et al., 2022) or a Transformer (Peebles & Xie, 2023; Bao et al., 2023a), is trained to iteratively denoise a random input, step-by-step, to generate a clean sample.

The Multimodal Diffusion Transformer (MMDiT) (Blattmann et al., 2023) departs from causal, autoregressive models (Esser et al., 2021; Yu et al., 2022) by using bidirectional attention, making it ideal for tasks requiring global compositional planning. This principle of modular control underpins flexible frameworks like Unidiffuser (Bao et al., 2023b) and Uni-renderer (Chen et al., 2025). By assigning independent noising and denoising schedules to each modality, these models enable a versatile "any-to-any" generation paradigm, where any subset of modalities can condition the synthesis of the rest.

### 2.2 CHINESE CALLIGRAPHY GENERATION

Early studies in Chinese calligraphy generation, whether based on GANs (Wu et al., 2020; Xie et al., 2021; Tang & Lian, 2021) or high-fidelity diffusion models (Zhang et al., 2024; Liu & Lian, 2024; He et al., 2024; Dai et al., 2024), have primarily treated the task as one-shot or few-shot font generation. These methods excel at synthesizing isolated characters with accurate structure (Zeng et al., 2023) but do not address page-level composition.

More recent efforts have shifted focus to page-level compositional synthesis, but these approaches exhibit critical limitations. On one hand, general-purpose image generation models and VLM-based systems can render full compositions but often fail on correctness, producing characters with improper forms or styles (Bai et al., 2025; OpenAI, 2024; Gong et al., 2025). On the other hand, specialized models like CalliPaint (Liao et al., 2023), Moyun (Liu et al., 2024), and CalliffusionV2 (Liao et al., 2024) generate characters sequentially. This autoregressive or sequential nature prevents holistic planning, leading to deficiencies in the global layout, rhythm, and inter-character ligatures that define a finished piece. In contrast, our work employs a non-autoregressive framework that plans the entire layout jointly, enabling a globally coherent composition that is both stylistically consistent and structurally accurate.

### 2.3 CHINESE CALLIGRAPHY RECOGNITION

The field of calligraphy recognition has evolved from analyzing isolated characters to employing end-to-end sequence-to-sequence models that handle connected and cursive scripts. A classic deep learning paradigm is the Convolutional Recurrent Neural Network (CRNN) (Shi et al., 2016), which combines a CNN feature extractor with an RNN decoder. More recently, the domain has been dominated by more powerful Transformer-based architectures (Dosovitskiy et al., 2021). State-of-the-art models, such as OracleNet (Zhou et al., 2025) and CalliReader (Luo et al., 2025), leverage these modern backbones to achieve high accuracy on stylistically diverse and irregular layouts. However, these highly effective methods are task-specialized recognizers. They operate independently of the

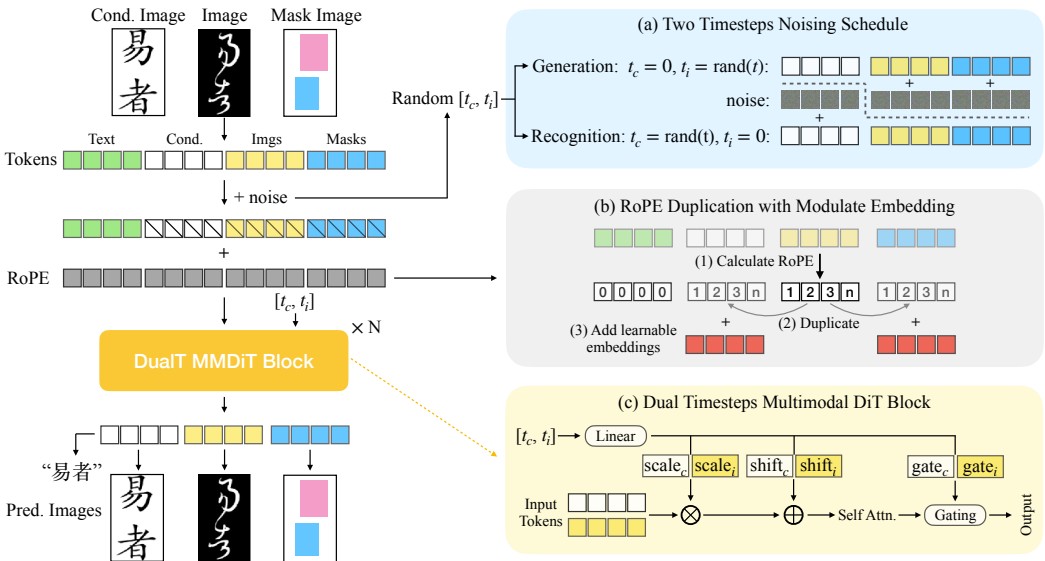

Figure 2: An overview of the UniCalli framework. Abbreviations are as follows: Cond. (Condition), Pred. (Prediction), Attn. (Attention), and RoPE (Rotary Positional Embedding).

generative process and lack a shared representation, a critical gap our unified framework directly addresses by integrating both tasks.

## 3 METHOD

This section details the methodology of our proposed framework, UniCalli, with an overview of the complete pipeline provided in Figure 2. We begin in Section 3.1 by introducing the core principle of our approach: a unified framework that treats calligraphy generation and recognition as mutually enhancing dual tasks. Following this, Section 3.2 delves into the specific architectural innovations for column-level modeling, including the use of a box map latent and our Duplicate RoPE strategy to fuse spatial information. Subsequently, Section 3.3 presents our Conditional Dropout technique, designed to disentangle style and glyph information and mitigate style overfitting. Finally, Section 3.4 describes our joint training scheme, which leverages a combination of labeled, unlabeled, and synthetic data to enhance the model's overall generalization capabilities.

### 3.1 UNIFIED FRAMEWORK FOR BIDIRECTIONAL LEARNING

We begin by defining our notation. Let the content image (a standard-font rendering), the calligraphy image, and its corresponding bounding box map be denoted by $I_c \in \mathbb{R}^{3 \times H \times W}$, $I_i \in \mathbb{R}^{3 \times H \times W}$, and $I_m \in \mathbb{R}^{3 \times H \times W}$, respectively. These images are first projected into a latent space using a pretrained Variational Autoencoder (VAE) encoder, denoted as $\mathcal{E}$. This yields the latent representations $z_c = \mathcal{E}(I_c)$, $z_i = \mathcal{E}(I_i)$, and $z_m = \mathcal{E}(I_m)$. We employ two independent timesteps, $t_c, t_i \in [0, 1]$, where $t_c$ controls the noising process for the content latent $z_c$, and $t_i$ governs the noising applied jointly to the calligraphy latent $z_i$ and the box map latent $z_m$.

At the heart of UniCalli is the principle that calligraphy generation and recognition are dual tasks that can mutually enhance one another. A model proficient in generating a character's visual form from its abstract identity should inherently possess the features needed to recognize that character from its image, and vice-versa. By training these two capabilities within a single, unified model, we enable the sharing of representations, forcing the model to learn a more robust and holistic understanding of the relationship between text, style, and spatial layout. This synergy is the core motivation for our unified design.

Our framework actualizes this principle through a training procedure that operates in one of two modes: **generation** or **recognition**, selected randomly at each training step. The entire process is

built upon the latent representations $z_c$, $z_i$, and $z_m$. We corrupt these latents using the flow-matching technique (Lipman et al., 2022). For a given latent $z_k$ and timestep $t_k \in [0, 1]$, the noised latent is $z_k^\epsilon = t_k \cdot \epsilon_k + (1 - t_k) \cdot z_k$, where $\epsilon \sim \mathcal{N}(0, I)$. The training mode dictates the timestep assignments. For **generation**, we aim to create an image and layout from content, so we set the content timestep $t_c = 0$ (no noise, used as condition) and sample the image timestep $t_i$ uniformly from $[0, 1]$. For **recognition**, the goal is to infer content from an image, so we set $t_i = 0$ and sample $t_c$ from $[0, 1]$.

This dual-mode approach is mirrored in our composite loss function. Let $L_{\text{cond}}$, $L_{\text{img}}$, and $L_{\text{box}}$ be the standard flow-matching losses for the latents. The total loss $L_{\text{total}}$ is conditioned on the training mode. In **generation mode**, the objective is to reconstruct the image and box map:

$$L_{\text{total}} = L_{\text{img}} + L_{\text{box}} + \lambda \cdot L_{\text{cond}}.$$

Conversely, in **recognition mode**, the objective is to reconstruct the content:

$$L_{\text{total}} = L_{\text{cond}} + \lambda \cdot (L_{\text{img}} + L_{\text{box}}).$$

Here, $\lambda$ is a balancing hyperparameter empirically set to $0.02$. This dual-objective strategy steers the unified model to learn the complete bidirectional mapping between content and its visual rendering.

## 3.2 Architecture for Column-Level Modeling

While the unified framework provides the learning structure, achieving high-fidelity **column-level** calligraphy requires specific architectural designs that can master the intricate spatial relationships between characters. We introduce two key innovations to address this.

First, to provide the model with explicit guidance on spatial layout, we incorporate a **box map latent** ($z_m$). This latent representation encodes the precise bounding box (position and scale) of each character within the column. By tasking the model with predicting this map during generation, we force it to learn the core principles of calligraphic composition. This explicit supervision of the column's structure is fundamental for rendering complex details accurately, such as realistic character spacing, size variations, and natural-looking ligatures that connect adjacent characters.

Second, for the model to effectively utilize the box map, the spatial information across all three modalities—content ($z_c$), image ($z_i$), and box map ($z_m$)—must be seamlessly fused. To achieve this, we propose a **Duplicate RoPE with Modulated Embedding** strategy. This technique establishes a shared spatial coordinate system for all modalities. We first compute the Rotary Position Embedding (RoPE) (Su et al., 2024) for the calligraphy image latent $z_i$. This RoPE, which contains rich 2D positional information, is then replicated and applied to both the content and box map latents. To allow each modality to retain its unique identity within this shared system, we add a distinct, learnable modulation embedding ($E_{\text{mod}}$) to each copy:

$$\text{RoPE}_i = \text{CalculateRoPE}(z_i), \tag{1}$$

$$\text{RoPE}_k = \text{RoPE}_i + E_{\text{mod\_k}}, \quad \text{for } k \in \{c, m\}. \tag{2}$$

This shared-yet-distinct representation is the key that enables the model to build strong bidirectional associations, allowing it to, for example, determine a character's appearance based on its identity and position, or infer its identity based on its appearance and position.

Finally, these spatially-aware latents are processed using an adapted MMDiT (Blattmann et al., 2023) block. The input tokens for each modality are modulated by their respective timesteps ($T_c = t_c$, $T_i = T_m = t_i$) before being concatenated and passed through a shared self-attention layer. This allows the model to integrate information from all modalities, conditioned on their individual states in the diffusion process, within a single, powerful block.

## 3.3 Disentangling Style and Glyph via Conditional Dropout

Standard conditional training often leads to overfitting on long-tail stylistic data, where the model prioritizes rare style constraints at the expense of correct glyph structure. To mitigate this, we employ a strategy to disentangle style representation from glyph formation via conditional dropout. Specifically, we mask the content condition with pure noise at a probability $p_{\text{drop}}$ by setting the condition timestep $t_c$ to 1:

$$t_c = \begin{cases} 1 & \text{with probability } p_{\text{drop}} \\ 0 & \text{with probability } 1 - p_{\text{drop}} \end{cases}. \tag{3}$$

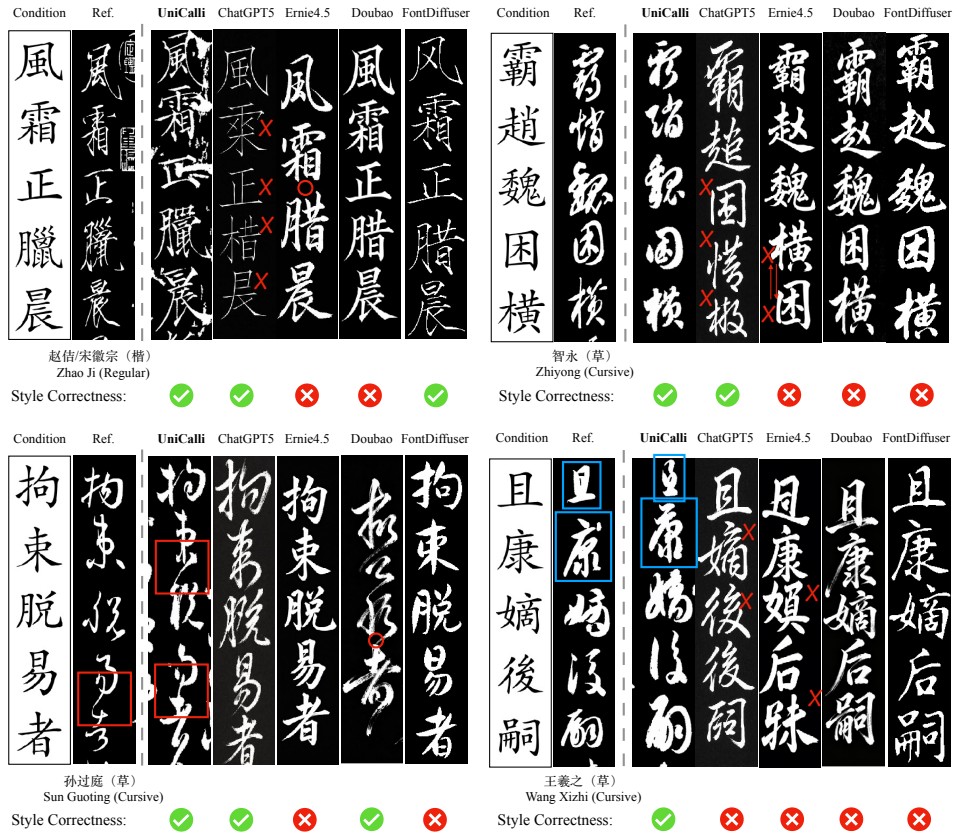

Figure 3: Qualitative comparison of our model against state-of-the-art generative models. To facilitate analysis, we use the following visual annotations: a red cross ✗ marks incorrectly rendered glyphs, while a red circle ⭕ indicates omitted characters. Furthermore, we highlight desirable calligraphic features: red boxes ❏ showcase well-formed connecting strokes, and blue boxes ❏ emphasize appropriate, context-aware character sizing. Beneath each generated image, we provide style correctness to evaluate its stylistic fidelity.

Empirically, a dropout rate of $p_{\text{drop}} = 0.05$ proves most effective. We observe that excessive dropout yields generic, style-agnostic outputs, while insufficient dropout leads to style over-prioritization, resulting in abstract patterns with incoherent glyph structures.

## 3.4 JOINT TRAINING ON LABELED, UNLABELED, AND SYNTHETIC DATA

The strategy of stochastically replacing the condition with noise, as detailed in the preceding section, can be framed as a form of unconditional generation. This perspective offers a natural mechanism for incorporating unlabeled data into our training paradigm. Specifically, data samples lacking annotations are processed by setting the condition timestep $t_c = 1$, which forces the conditional latent into a pure noise state, $z_c^\epsilon = \epsilon_c$. This technique seamlessly integrates unlabeled data by treating its generation as an unconditional task, thereby enriching the model's understanding of diverse calligraphic styles.

Furthermore, to expand the model's glyph repertoire and improve its structural understanding of characters, we incorporate a large corpus of synthetic data. We curated a collection of TrueType Font (TTF) files for various script styles, including Regular/Kai, Running/Xing, and Cursive/Cao. These fonts were employed to render extensive content from both ancient and modern Chinese literature. The joint training on synthetic data significantly broadens the model's known character set, while the unlabeled data enhances its grasp of calligraphic styles. Collectively, these heterogeneous data sources substantially improve the model's overall generalization capabilities.

# 4 EXPERIMENTATION

In this section, we detail our experimental setup, including the dataset and model architecture. We then present a comprehensive evaluation of our model, UniCalli. In Section 4.1, we benchmark UniCalli's generation and recognition capabilities against several state-of-the-art models. To validate our design choices, we conduct a series of ablation studies in Section 4.2. Finally, to assess the model's robustness and generalizability, we extend our evaluation to the challenging domains of ancient scripts in Section 4.3.

**Dataset.** Our experimental dataset was constructed from multiple sources. We first curated a collection of over 8,000 Chinese calligraphy images (examples can be viewed at Appendix C), featuring the works of 93 calligraphers across five major script styles: Regular/Kai, Running/Xing, Cursive/Cao, Clerical/Li, and Seal/Zhuan. From this collection, a subset of over 4,000 images was annotated, yielding more than 150,000 character instances, each with a specified bounding box and character label.

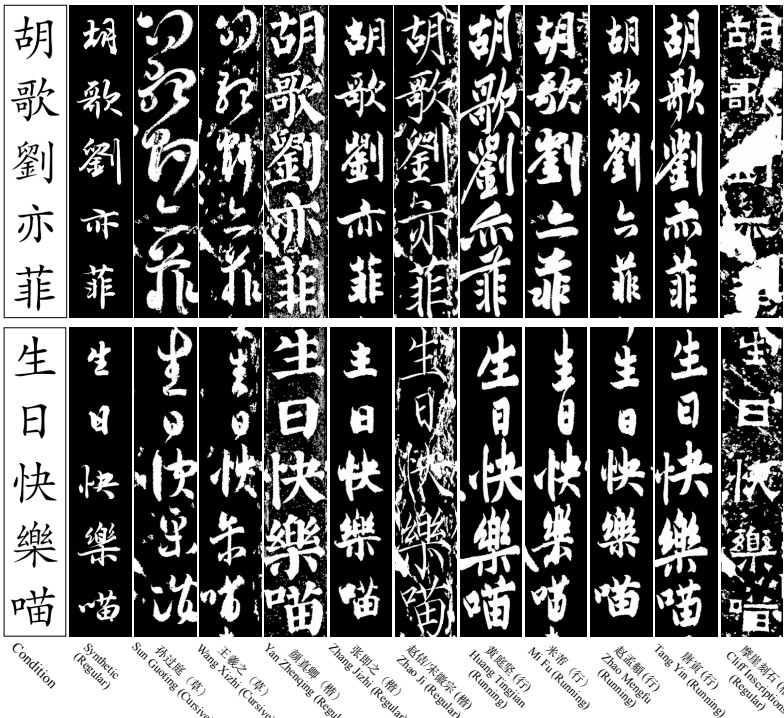

Figure 4: Demonstration of UniCalli's robust multi-style calligraphy generation. The same content is rendered in the distinct styles of various calligraphers. The top panel showcases the generation of two Chinese celebrity names, while the bottom panel displays a birthday greeting. This versatility highlights UniCalli's robustness in capturing diverse calligraphic styles and its strong potential for a wide range of downstream applications.

**Model.** We fine-tune the **FLUX** (Labs et al., 2025) backbone with our **Duplicate RoPE with Modulated Embedding** strategy. The model's input consists of three images: a standard-font content image, a calligraphy style reference, and a bounding box map. For each sample, we crop a region of five consecutive characters and resize both the crop and its box map to $128 \times 640$. The content image is formed by horizontally concatenating five $128 \times 128$ standard-font renderings. All three $128 \times 640$ images are then patchified and concatenated before being input to the model. We fine-tune for 500k iterations on 8×H100 GPUs.

## 4.1 COMPARISONS

**Generation.** We benchmark UniCalli's generation capabilities against one-shot font generation method: FontDiffuser (Yang et al., 2024), VLM-based models: ChatGPT-5 (OpenAI, 2024), Ernie-

4.5-Turbo (Research, 2025), and Doubao (Gong et al., 2025) in two settings. For reference-based synthesis, where a ground-truth image exists, visual comparisons are shown in Figure 3, and quantitative metrics (L1, SSIM, LPIPS, FID) are reported in Table 3. For reference-free synthesis, Figure 4 demonstrates UniCalli's ability to generate a character in diverse styles given the same content. Finally, we conducted a user study (details in Appendix H) with 20 calligraphy enthusiasts who ranked the outputs based on Style Consistency, Glyph Accuracy, Naturalness, and Overall Preference, with results summarized in Table 1.

Table 1: Unified user study results for reference-free calligraphy generation. We report the **(mean, standard deviation)** of user ratings on a 5-point Likert scale (1=Worst, 5=Best) across four key metrics. The metrics are: Style Fidelity, Glyph Accuracy, Naturalness, and Overall Preference. The best score in each category is highlighted in **bold**. The arrow (↑) indicates that higher scores are better.

| Method | Style Fidelity↑ | Glyph Accuracy↑ | Naturalness↑ | Overall Preference↑ |
|---|---|---|---|---|
| FontDiffuser | 1.680, 1.420 | **4.950, 0.380** | 2.120, 1.550 | 1.890, 1.380 |
| ChatGPT-5 | 2.987, 1.205 | 3.853, 1.163 | 2.373, 1.334 | 2.373, 1.220 |
| Ernie-4.5 | 2.000, 1.166 | 3.560, 1.369 | 2.533, 1.226 | 2.507, 1.204 |
| Doubao | 2.413, 1.234 | 4.800, 0.462 | 3.520, 1.290 | 3.933, 0.573 |
| **UniCalli** | **4.267, 0.943** | 4.827, 0.443 | **4.520, 0.755** | **4.560, 0.787** |

Table 2: Character-level recognition accuracy on the held-out test set; **bold** indicates the best per row. "Doubao-1.5*", "Ernie-4.5*", and "Qwen-2.5*" denote Doubao-1.5-Thinking-Vision-Pro, Ernie-4.5-Turbo-VL, and Qwen-2.5-VL-7B, respectively.

| Category | GPT-4o | Ernie-4.5* | Qwen-2.5* | GOT-OCR2.0 | PP-OCRv5 | Doubao-1.5* | CalliReader | **Ours** |
|---|---|---|---|---|---|---|---|---|
| Cursive/Cao | 0.073 | **0.255** | 0.127 | 0.091 | 0.091 | 0.200 | 0.164 | 0.109 |
| Regular/Kai | 0.502 | 0.616 | 0.600 | 0.314 | 0.396 | 0.588 | 0.600 | **0.688** |
| Clerical/Li | 0.293 | 0.453 | 0.453 | 0.187 | 0.160 | 0.507 | 0.507 | **0.518** |
| Running/Xing | 0.364 | 0.600 | 0.473 | 0.336 | 0.436 | 0.545 | **0.658** | 0.528 |
| Seal/Zhuan | 0.067 | **0.133** | 0.067 | 0.000 | 0.000 | **0.133** | 0.067 | 0.050 |
| Total | 0.380 | 0.534 | 0.482 | 0.266 | 0.324 | 0.510 | 0.533 | **0.540** |

**Recognition.** We benchmark our recognizer against six models, including GPT-4o (OpenAI, 2024) and others. From the labeled data, we created 100-image validation and test sets. All models were evaluated on this common test set, and our model was selected based on validation performance. As shown in Table 2, our method attains the highest overall character-level accuracy across five script styles. We acknowledge that this advantage may be partly due to the similar distributions of our training and test data.

## 4.2 ABLATION STUDIES

**Components ablation.** We systematically evaluate the impact of the three main components of our framework (Table 4): (1) the Duplicate RoPE with Modulated Embedding, (2) the Joint Training Strategy utilizing synthetic, labeled, and unlabeled data, and (3) the Conditional Dropout mechanism. To establish a rigorous comparison, we define the **Baseline** as a standard FLUX Labs et al. (2025) model trained exclusively on the generation task. In this baseline setup, the clean condition, noisy image, and mask are concatenated into a single input sequence processed by a single, continuous RoPE, without our proposed structural decoupling or joint optimization.

**Value of $p_{\text{drop}}$.** The $p_{\text{drop}}$ (Eq. 3) value is the primary hyperparameter in our Conditional Dropout mechanism. It influences the model's focus, creating a trade-off between learning the fundamental **glyph structure** and capturing the specific calligraphic **style**. A higher $p_{\text{drop}}$ value encourages the model to ignore stylistic information and concentrate on the core character shape, while a lower value allows for more stylistic detail to be preserved, as shown in Figure 5.

## 4.3 ROBUSTNESS ON ANCIENT CHARACTERS.

**Oracle bone script.** We trained our model on the HUST-OBC dataset (Wang et al., 2024) and structured our evaluation into three distinct tasks. First, we assessed the model's ability to generate

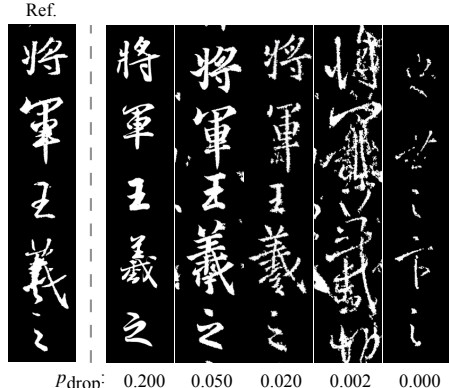

$p_{\text{drop}}$: 0.200  0.050  0.020  0.002  0.000

Figure 5: Ablation study on $p_{\text{drop}}$. An excessively low $p_{\text{drop}}$ causes the model to sacrifice structural integrity to replicate long-tail styles, whereas an excessively high value leads to style-agnostic, canonical characters due to over-disentanglement.

Table 3: Quantitative comparison with state-of-the-art methods on reference-based synthesis. The best score in each category is highlighted in **bold**.

| Method | L1↓ | SSIM↑ | LPIPS↓ | FID↓ |
|---|---|---|---|---|
| FontDiffuser | 0.370 | 0.425 | 0.527 | 80.25 |
| ChatGPT-5 | 0.201 | 0.331 | 0.412 | 55.50 |
| Ernie-4.5 | 0.375 | 0.507 | 0.475 | 68.38 |
| Doubao | 0.229 | 0.463 | 0.456 | 47.26 |
| **UniCalli** | **0.152** | **0.602** | **0.313** | **37.69** |

Table 4: Ablation study of our model's key components. Improved LPIPS and FID indicate enhanced diversity and realism, avoiding the mode collapse associated with pixel-wise averaging.

| Metric | L1↓ | SSIM↑ | LPIPS↓ | FID↓ |
|---|---|---|---|---|
| Baseline | 0.200 | 0.551 | 0.430 | 52.90 |
| + Joint Training | 0.160 | 0.604 | 0.387 | 46.42 |
| + RoPE Duplication | **0.148** | **0.613** | 0.352 | 41.78 |
| + Cond. Dropout | 0.152 | 0.602 | **0.313** | **37.69** |

Table 5: Quantitative comparison of UniCalli and OracleNet on Oracle Bone Script tasks. Generation accuracy is evaluated by human experts and classified into three categories: Completely Correct, Largely Correct, and Completely Incorrect (detailed evaluation criteria are shown in Appendix G). Recognition accuracy is measured against deciphered ground truth. N/A denotes that a method is not applicable to the task.

| Method | Generation Accuracy (%) | | | Recognition Acc. (%) |
|---|---|---|---|---|
| | Completely Correct | Largely Correct | Completely Incorrect | |
| OracleNet | | N/A | | **73.9%** |
| **UniCalli** | **67%** | **20%** | **13%** | 62.5% |

oracle bone script characters from corresponding modern Chinese characters. Second, we evaluated its performance on supervised oracle bone character recognition, using the Top-K accuracy metric to quantify precision. For the third task, we applied our model to a set of 100 undeciphered oracle bone characters to predict their potential modern Chinese character equivalents. The qualitative aspects of our study—namely, the generative task were evaluated by experts in oracle bone script from the School of History at Xiamen University, details in Appendix G, visual results are shown in Figure 10. Furthermore, we benchmarked our model against OracleNet (Zhou et al., 2025) on a curated set of 100 deciphered characters, as detailed in Table 5. Recognition accuracy was calculated directly, while generation correctness was assessed by experts.

**Egyptian hieroglyphs.** We conducted experiments on dataset (Umer, 2023) (see Appendix F for more details).

# 5 CONCLUSION

We introduced UniCalli, a unified diffusion framework that advances computational Chinese calligraphy by unifying isolated character generation with holistic, page-level composition. Our contributions include a large-scale, annotated dataset and a novel Multimodal Diffusion Transformer that jointly handles generation and recognition with global coherence. Trained on diverse data, UniCalli faithfully captures the stylistic nuances of master calligraphers, including complex ligatures and spatial rhythms, while maintaining strict glyph accuracy. The framework's robustness is shown by its successful extension to other ancient writing systems like Oracle bone script and Egyptian hieroglyphs, highlighting its potential as a powerful tool for the digital preservation and scholarly study of global cultural heritage.

## 6 ETHICS STATEMENT

This research was conducted with the aim of preserving and promoting cultural heritage through computational methods. The dataset was created from digitized historical works, many of which are in the public domain, and augmented with synthetically generated data from publicly available fonts and literature. We believe this work has a positive cultural impact by making the art of calligraphy more accessible. The code and dataset will be released publicly to encourage further academic research and creative applications in a responsible manner.

## 7 REPRODUCIBILITY STATEMENT

We are committed to ensuring the reproducibility of our research. This paper provides a detailed description of our framework, UniCalli, the data collection and annotation process, and our experimental setup in Sections 3 and 4. To facilitate the verification of our results and to allow other researchers to build upon our work, we will make our source code, the curated dataset with annotations, and the pre-trained model weights publicly available upon publication, as stated in the abstract.

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

## A  WRITING ASSISTANCE (LLM USE DISCLOSURE)

We utilized a large language model (LLM), specifically ChatGPT, as a writing assistant to enhance the quality of this manuscript. The tool's role was strictly limited to improving language, including grammar, clarity, and academic tone. All scientific content—including the core ideas, methodology, analyses, and experimental results—was generated exclusively by the authors. We have carefully reviewed every modification suggested by the LLM to ensure it aligns with our original intent and maintains factual accuracy. The authors retain full responsibility for the final content of this paper.

## B  LIMITATIONS

The limitations of this work are twofold. Firstly, the historical calligraphy data contains considerable noise from age and poor preservation, which persists in the generated outputs despite our denoising efforts. Secondly, the severe long-tail distribution of the data, caused by the rarity of works from some calligraphers, is difficult to optimize algorithmically and results in deviations from the original artistic styles.

## C  DATA PRE-PROCESSING AND EXAMPLES

To simplify the learning task, all calligraphy images were preprocessed through denoising and binarization, and subsequently categorized by their background color (black or white). To augment our dataset, we generated synthetic data using three TrueType Font (TTF) files for Regular, Running, and Cursive scripts, rendering text from a large corpus of classical and modern Chinese literature (e.g., Dream of the Red Chamber, Water Margin, and the collected works of Lu Xun). To ensure the model prioritized learning from authentic works, the sampling probability for synthetic data was set to 0.2 during training. Our conditioning prompts were structured into four parts: data source (real or synthetic), background color, calligrapher description, and script style description.

### C.1  REAL IMAGE PROCESSING

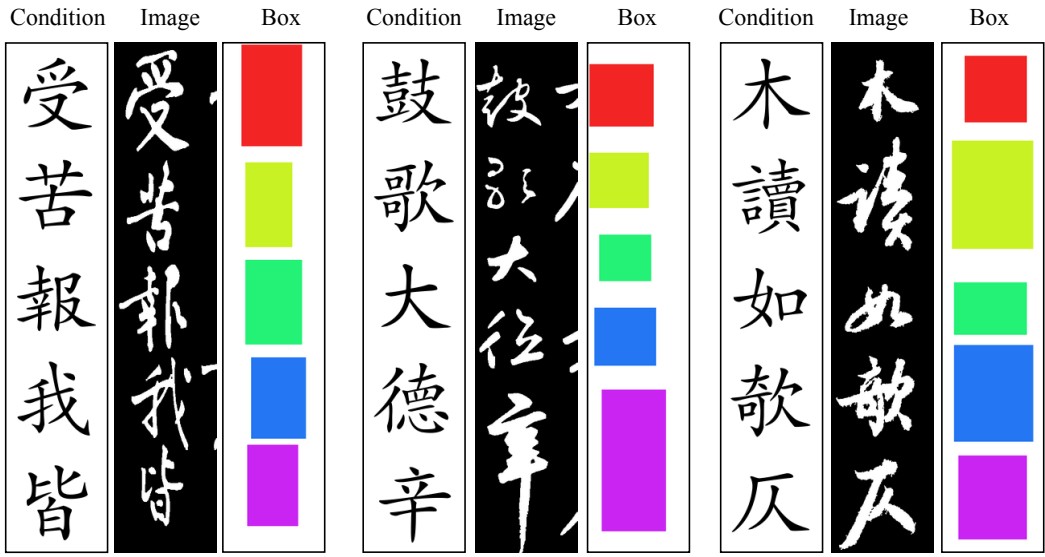

Figure 6: Labeled data examples.

This stage processes high-resolution scans of historical calligraphy, which are accompanied by annotations detailing the location and identity of each character.

**Vertical Text Segment Extraction:** Chinese calligraphy is traditionally written in vertical columns. We first group the annotated characters into columns by clustering them based on their horizontal

coordinates. To create training samples of a consistent length, we filter for columns containing at least a minimum number of characters, $N$ (e.g., $N = 5$). From these valid columns, we randomly sample a continuous vertical sequence of $N$ characters. This process acts as a form of random cropping, increasing data diversity.

**Image Cropping and Augmentation:** A tight bounding box is calculated around the selected character segment, and the original image is cropped to this region with a small padding margin. As an optional data augmentation step, we employ an automatic binarization algorithm. This algorithm determines the image's polarity (light text on a dark background or vice-versa) by generating two candidate binary masks and scoring them based on two criteria:

- **Foreground Ratio:** How well the proportion of foreground pixels falls within a predefined, typical range for text.

- **Edge Contrast:** The degree to which Canny edges align with the interior of the foreground mask, indicating sharp, well-defined strokes. The higher-scoring mask is chosen, normalizing the image to white text on a black background. This makes the model robust to variations in paper and ink.

## C.2 Synthetic Data Generation

To augment our dataset and provide the model with a wider variety of styles and characters, we synthesize additional training samples.

**Text and Style Sampling:** A text segment of a random length (up to $N$) is sampled from a large corpus of Chinese literature. Concurrently, a TrueType Font (TTF) is randomly selected from a curated collection of diverse calligraphic fonts.

**Image Rendering:** The sampled text is rendered onto a blank canvas using the chosen font. The background and text colors (either black-on-white or white-on-black) are also selected randomly to ensure variability.

## C.3 Conditioning Signal Formulation

A key aspect of our pipeline is the explicit separation of content and style into distinct conditioning signals.

**Content Conditioning:** For every sample, whether real or synthetic, we generate a **content image**. This is a standardized representation where the target sequence of $N$ characters is rendered in a single, consistent, non-stylized font (e.g., Regular/Kai) at fixed positions on a white background. This image serves as an unambiguous guide for "what" characters the model should generate, isolating content from stylistic attributes.

**Style Conditioning:** A descriptive **text prompt** is constructed to guide the artistic style. For real images, this prompt includes metadata such as the script type (e.g., Cursive, Seal), the author's name, and pre-defined stylistic descriptions associated with that author or script. For synthetic data, the prompt includes the name of the source TTF font and its associated style descriptors. All Chinese terms within the prompt are converted to Pinyin to maintain a consistent vocabulary.

Finally, all images (target, content) are resized to a fixed resolution (e.g., $128 \times 640$) and normalized to a pixel value range of $[-1, 1]$. The text prompt and the character sequence are tokenized for model consumption. The final output for each training step is a tuple containing the target image, the content-conditioning image, the style-conditioning prompt, and the tokenized character IDs.

To prepare our model inputs, we process segments from the original source data through cropping and labeling. We first crop the data to isolate relevant regions of interest, as shown in the source examples in Figure 7. Each cropped segment is then annotated with a ground-truth label. This creates the final curated dataset of labeled inputs, ready for model training. Examples of these final, labeled inputs are provided in Figure 6.

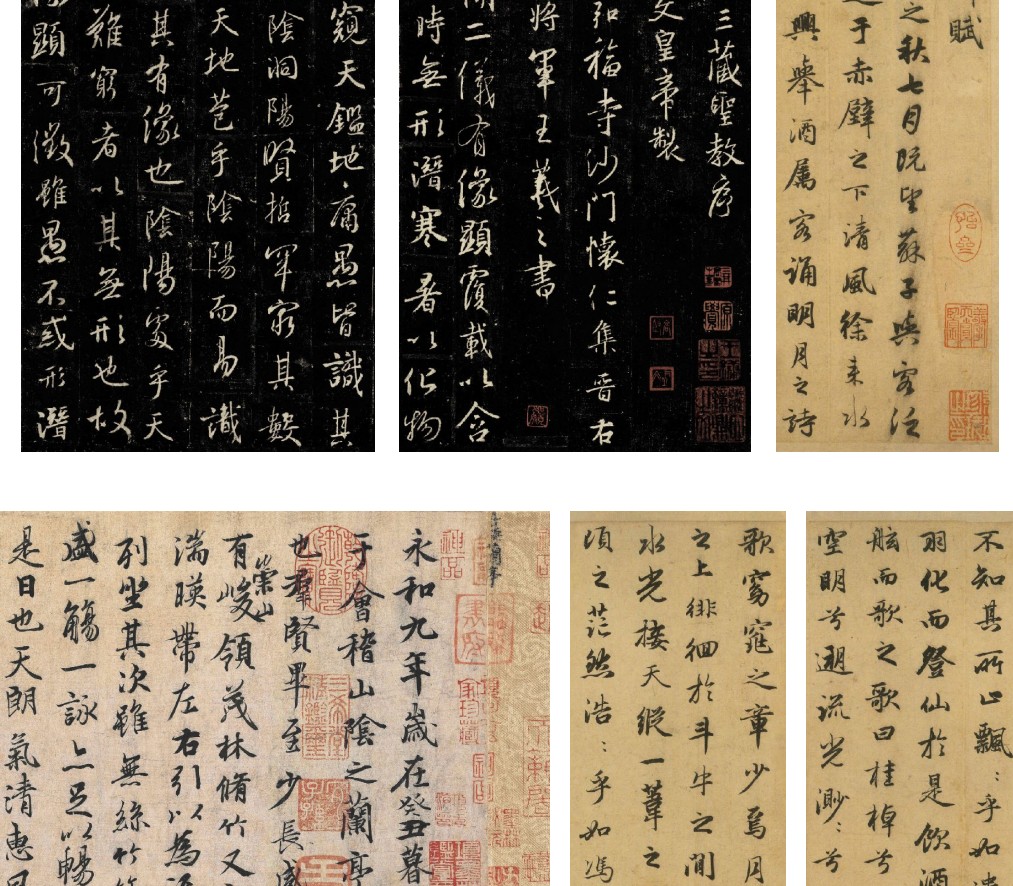

Figure 7: Dataset example.

## C.4 ARTIFACTS IN GENERATED SAMPLES

**Origin of Visual Anomalies.** A qualitative examination of certain generated samples (e.g., styles corresponding to Sun Guoting and Yan Zhenqing) reveals the presence of irregular "white patches" or discontinuities within the stroke glyphs. We wish to clarify that these are not generative hallucinations or model failures, but rather high-fidelity reproductions of artifacts inherent to the training dataset.

**Physical and Digital Causes.** A significant portion of the training corpus is derived from historical stone stele rubbings (beike). These physical monuments have been subjected to centuries of natural weathering, erosion, and mechanical damage, resulting in surface pitting and cracks . When these historical rubbings undergo digital binarization for dataset curation, these physical imperfections—where the paper could not make contact with the eroded stone surface—are converted into binary "noise." Consequently, the model, which aims to approximate the underlying data distribution, learns to faithfully reconstruct these weathering artifacts as part of the stylistic texture.

**Future Mitigation.** We acknowledge that while historically authentic, these artifacts may detract from the aesthetic cohesion of the generated calligraphy for certain applications. In future work, we plan to implement rigorous data-level preprocessing, utilizing dedicated denoising algorithms to restore the integrity of the glyph strokes before training, thereby separating the calligraphic content from the historical degradation of the substrate.

# D EVALUATION ON PUBLIC BENCHMARKS

To ensure a fair and robust evaluation of our recognizer, we extended our experiments to include third-party public datasets. However, it is important to note that, to the best of our knowledge, there are currently no publicly available datasets specifically dedicated to page-level Ancient Chinese Calligraphy. Existing public datasets in this domain primarily focus on isolated, individual characters Zhao et al. (2025).As the closest available alternative, we selected CalliBench Luo et al. (2025), a recently released benchmark for page-level calligraphy. It is crucial to emphasize that while CalliBench supports page-level evaluation, its data distribution differs significantly from our training objective: it predominantly consists of modern artistic fonts and contemporary calligraphy, rather than the historical ancient styles (e.g., Yan style, Cursive script) that UniCalli is optimized for. This introduces a substantial domain gap between the training and testing distributions.Despite this challenge, we evaluated UniCalli on the CalliBench "Easy" subset against state-of-the-art general-purpose Vision-Language Models (VLMs) and specialized OCR systems. The results are reported in Table 6.

As shown in Table 6, UniCalli achieves a competitive F1 score of 0.498. While specialized OCR models like GOT-OCR2.0 achieve higher performance—likely due to their extensive training on broad, modern font datasets—UniCalli notably outperforms powerful general-purpose VLMs such as GPT-4o (F1: 0.400) and established industrial OCR systems like PP-OCRv5 (F1: 0.205). This result is encouraging: it demonstrates that although UniCalli was trained strictly on ancient historical data, it has learned robust, generalized features that allow for effective zero-shot transfer to unseen modern artistic styles.

Table 6: Quantitative comparison of recognition performance on the CalliBench Luo et al. (2025) (Easy subset). Note that UniCalli is trained exclusively on ancient historical calligraphy, whereas CalliBench consists largely of modern artistic fonts, introducing a domain gap.

| Model | Precision ↑ | Recall ↑ | F1 ↑ | NED ↓ |
|---|---|---|---|---|
| Ernie-4.5-Turbo-VL | 0.542 | 0.481 | 0.482 | **0.637** |
| Doubao-1.5-Thinking-Vision-Pro | 0.442 | 0.513 | 0.462 | 0.655 |
| PP-OCRv5 | 0.372 | 0.291 | 0.205 | 0.859 |
| GPT-4o | 0.457 | 0.403 | 0.400 | 0.726 |
| Qwen-2.5-VL-7B | 0.440 | **0.736** | 0.534 | 0.710 |
| GOT-OCR2.0 | **0.687** | 0.550 | **0.593** | 0.651 |
| **UniCalli** | 0.480 | 0.520 | 0.498 | 0.680 |

# E IMPLEMENTATION DETAILS OF UNIFIED ARCHITECTURE

## E.1 ZERO-SHOT TEXT RECOGNITION VIA FEATURE RETRIEVAL

Unlike traditional OCR systems that rely on autoregressive text decoding or classification heads, our unified framework performs recognition through a *zero-shot, retrieval-based mechanism*. This approach leverages the shared latent space between the generation and recognition tasks. The process consists of two phases:

1. **Reference Library Construction (Offline):**
   We construct a comprehensive reference feature library $\mathcal{L}$ prior to inference. We render images for all standard characters $c$ present in our vocabulary (derived from the standard `.ttf` font file used in training). Each rendered character image $I_c$ is passed through the

pre-trained VAE encoder $\mathcal{E}$ to obtain its latent feature representation $\mathbf{z}_c$:

$$\mathcal{L} = \{(c, \mathbf{z}_c) \mid \mathbf{z}_c = \mathcal{E}(I_c), \forall c \in \mathcal{V}\} \quad (4)$$

where $\mathcal{V}$ is the character vocabulary.

2. **Inference and Matching:**
   During the recognition task, the model outputs a predicted condition feature $\hat{\mathbf{z}}_{\text{pred}}$ (the tensor immediately preceding the VAE decoder). To determine the text content, we compute the cosine similarity between $\hat{\mathbf{z}}_{\text{pred}}$ and all reference features in $\mathcal{L}$. The recognized character $\hat{c}$ is determined by the nearest neighbor in this latent space:

$$\hat{c} = \underset{c \in \mathcal{V}}{\operatorname{argmax}} \left( \frac{\hat{\mathbf{z}}_{\text{pred}} \cdot \mathbf{z}_c}{\|\hat{\mathbf{z}}_{\text{pred}}\| \|\mathbf{z}_c\|} \right) \quad (5)$$

This design avoids the need for an auxiliary OCR module and ensures that the recognition capability is intrinsically aligned with the generative features of the model.

### E.2 DUPLICATE ROPE WITH MODULATED EMBEDDING

To process heterogeneous inputs (Image, Condition, and Mask) within a unified spatial context, we employ a **Duplicate RoPE** strategy augmented with **Learnable Modulated Embeddings**. This design ensures that the model can distinguish between input modalities without disrupting the spatial geometry defined by Rotary Positional Embeddings (RoPE).

**Mechanism.** Let $\mathbf{X} \in \mathbb{R}^{L \times d}$ represent the input sequence of features for a specific modality $m \in \{\text{Image}, \text{Condition}, \text{Mask}\}$. We initialize a unique, learnable embedding vector $\mathbf{v}_m \in \mathbb{R}^d$ for each modality, initialized to zeros. This vector is broadcasted across the sequence length and added element-wise to the input features before the attention projection:

$$\mathbf{X}'_m = \mathbf{X}_m + \mathbf{v}_m \quad (6)$$

Subsequently, the transformed features $\mathbf{X}'_m$ are projected into Query ($\mathbf{Q}$) and Key ($\mathbf{K}$) states, to which RoPE is applied.

**Preservation of Relative Positioning.** It is crucial to note that adding the modality embedding $\mathbf{v}_m$ does not interfere with the relative positional encoding properties of RoPE.

- **Spatial Structure:** RoPE injects positional information via rotation based on the token index $pos$.
- **Modality Identity:** The learnable embedding $\mathbf{v}_m$ acts as a global, modality-specific "offset" or bias in the semantic space.

By superimposing $\mathbf{v}_m$, we allow the model to distinguish *what* the input is (e.g., "this is a condition token" vs. "this is an image token") while RoPE simultaneously tells the model *where* the input is. Since the two signals operate orthogonally—one as a static semantic bias and the other as a dynamic frequency-based rotation—the underlying relative positional "grid" remains intact across all duplicated modalities.

## F EGYPTIAN HIEROGLYPHS.

Directly translating from English to Egyptian hieroglyphs is unfeasible due to their fundamentally different structures. English utilizes a concise alphabetic script, where a small set of abstract symbols represents phonemes (units of sound). In stark contrast, the ancient Egyptian system is vastly more complex, employing hundreds of signs that function as logograms (signs for words), phonograms (signs for sounds), and determinatives (semantic classifiers). A simple one-to-one phonetic or symbolic mapping between these systems is therefore impossible.

To solve this challenge, our approach uses Chinese script as a semantic bridge in a two-step process. The rationale is that both Chinese characters and Egyptian hieroglyphs share a logographic foundation, where symbols are often rooted in pictorial representations of concepts. This structural parallel

allows us to bypass direct phonetic transliteration in favor of a semantic-first approach, mapping the core meaning of an English word rather than its sound. The complete pipeline is illustrated in Figure 8.

By leveraging this intermediary, our method transforms an impossible phonetic transliteration into a feasible conceptual and iconographic mapping. This creates a more logical and meaningful bridge between the ancient and modern languages, preserving semantic intent.

To validate our framework, we conducted experiments on the Kaggle-Egyptian-Hieroglyphs dataset (Umer, 2023) to assess its performance both qualitatively and quantitatively. The qualitative results, presented in Figure 9, showcase the high visual fidelity and contextual relevance of the generated hieroglyphs from English inputs. Concurrently, for a quantitative measure, the model's strong recognition accuracy is summarized in Table 7, empirically confirming the method's effectiveness.

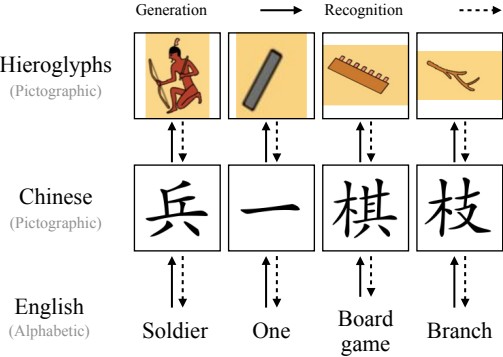

Figure 8: An illustration of the mapping process from English to Egyptian hieroglyphs. Direct mapping is challenging due to the fundamental differences between English, an alphabetic script, and Egyptian hieroglyphs, a pictographic script. To bridge this gap, our approach first maps English to Chinese—a script that is also fundamentally pictographic—which then serves as an intermediate representation for the final mapping to hieroglyphs.

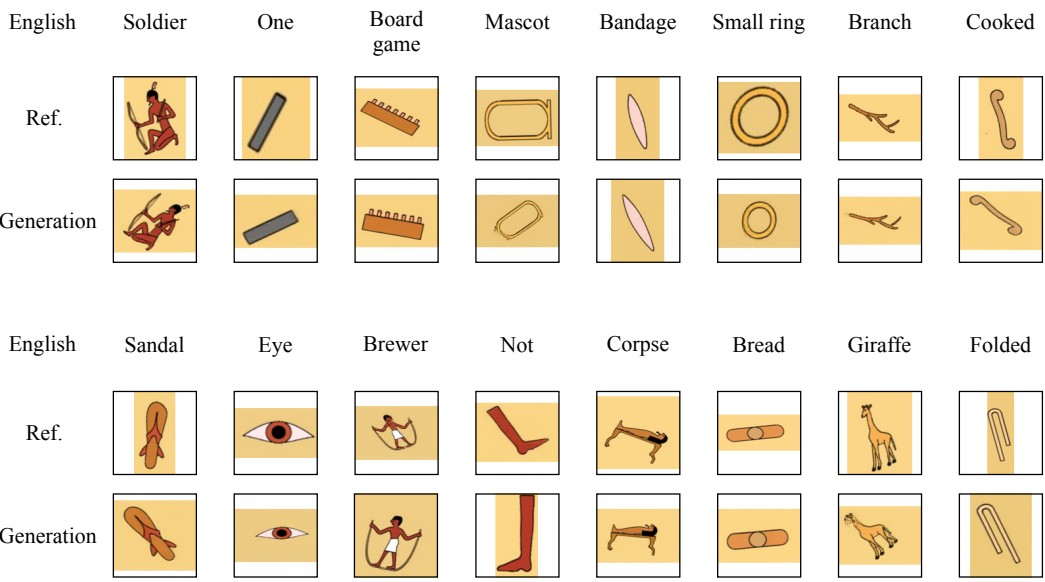

Figure 9: Generation results of Egyptian hieroglyphs. The first row shows the input text in English. The second row is the reference image, and the third row displays the generated output.

Table 7: Accuracy of Egyptian hieroglyphs recognition.

|  | Accuracy |
|---|---|
| UniCalli | 0.96 |

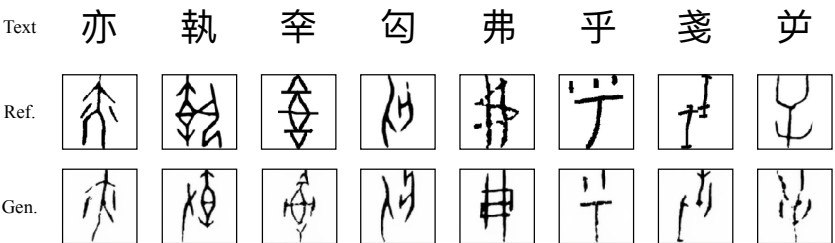

Figure 10: Qualitative results of our model for Oracle Bone Script generation. For each example, we show the input modern Chinese character (top row), a corresponding ground-truth glyph as reference (middle row), and the Oracle Bone Script generated by our method (bottom row). Our generated results are highly consistent with the reference images in both structure and style.

## G   HUMAN EXPERT EVALUATION CRITERIA FOR GENERATION

This section details the criteria used by human experts to evaluate the quality of the generated oracle bone script characters. Each generated character is assessed and categorized into one of three tiers based on its structural and stylistic accuracy compared to the ground truth. The scoring guidelines are as follows:

- **Completely Correct:** The generated character is structurally identical to the ground truth character or represents a well-accepted calligraphic variant. All strokes are correctly formed and placed.
- **Largely Correct:** The generated character captures the essential structure and is clearly recognizable, but contains minor inaccuracies. These may include incorrect stroke thickness, slight misplacement of components, or minor stylistic deviations that do not alter the character's identity.
- **Completely Incorrect:** The generated character is structurally flawed, unrecognizable as the target character, or resembles a different character entirely.

Figure 11 provides visual examples for each of these categories, illustrating the practical application of our scoring guidelines.

## H   USER STUDY OF GENERAL CALLIGRAPHY GENERATION

To evaluate the qualitative performance of our model, we designed a user study consisting of 10 questions. We recruited 20 participants, all of whom identified as calligraphy enthusiasts, to complete the survey.

Each question, or evaluation task, presented participants with three components:

- **Prompt:** A textual prompt specifying the content to be generated, a description of the target calligrapher's style, and the desired script style (e.g., Running Script, Clerical Script).
- **Reference Image:** An image containing an authentic work excerpt from the specified calligrapher to serve as a ground-truth style example.
- **Generated Images:** A set of calligraphy images generated by UniCalli and the baseline models for comparison. To prevent positional bias, the order of these images was randomized for each question and each participant.

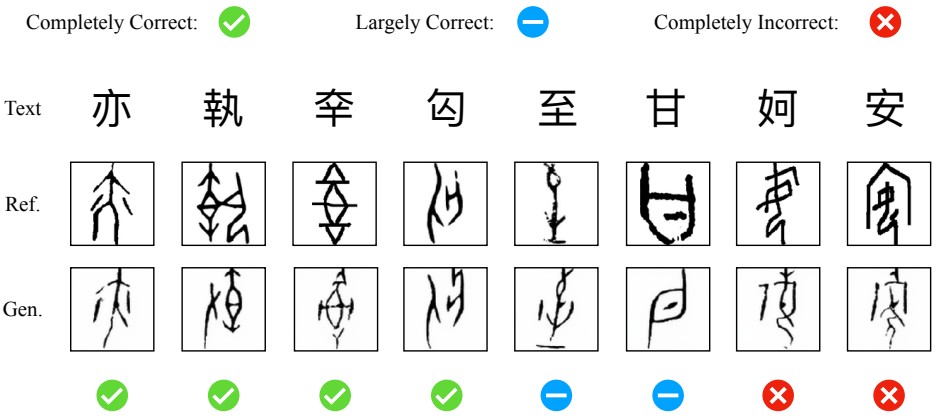

Figure 11: Examples illustrating the evaluation criteria for generated oracle bone script characters. Each row demonstrates samples categorized as (a) Completely Correct, (b) Largely Correct, and (c) Completely Incorrect by human experts.

Participants were asked to rate the generated images based on four key metrics: **Style Fidelity**, **Glyph Accuracy**, **Naturalness**, and **Overall Preference**. Ratings were provided on a 5-point Likert scale, where 1 corresponds to "Worst" and 5 to "Best". The aggregated results, reported in the main paper, represent the mean and standard deviation of these ratings. An example of the user study interface is shown in Figure 12.

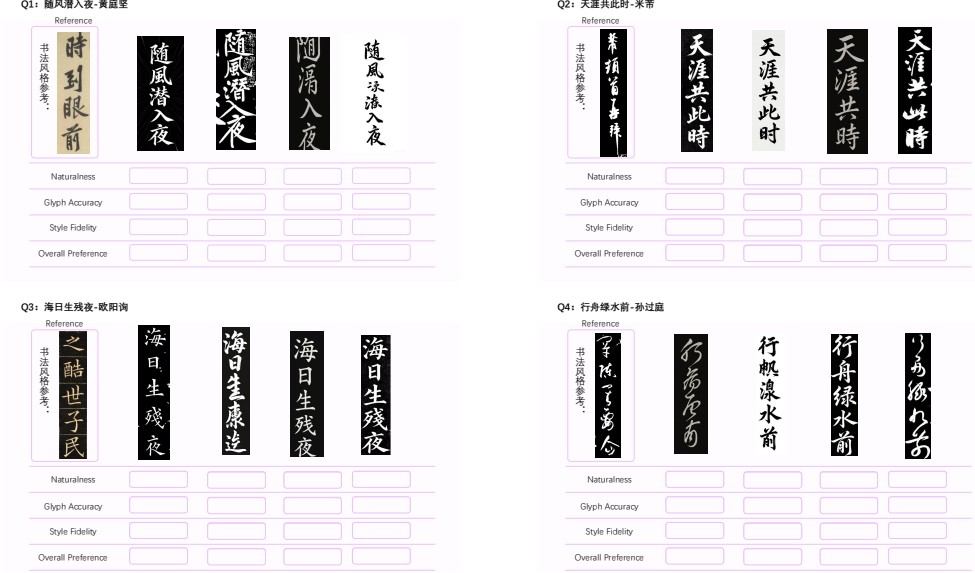

Figure 12: An example from our user study questionnaire.

### H.1 QUALITATIVE FEEDBACK

In addition to quantitative ratings, we collected qualitative feedback from the participants. Below are some representative comments, translated from the original responses:

- **Participant 1:** "The second option in Q1 (generated by UniCalli) is a very convincing replica. Huang Tingjian's style is often described as resembling a 'dead snake hanging from a tree.' The other options devolved into an unorthodox and unrefined 'Jianghu' style or were simply inaccurate."

- **Participant 2:** "Regarding Q2, the second option exhibits an unrefined 'Jianghu' style and incorrectly uses simplified characters, leading to a distorted result. The first option is similarly unrefined. While the third option shows some resemblance to the style, it lacks structural accuracy. In comparison, the fourth option (generated by UniCalli) is significantly better, though for running script, it could still improve the fluid connection and calligraphic 'echo' between adjacent characters."

- **Participant 3:** "Having practiced Yan Zhenqing's calligraphy myself, I found that option 2 in Q7 (generated by UniCalli) was a clear and striking match for his style."

- **Participant 4:** "In Q8, the 'Slender Gold' style, characteristic of Emperor Huizong's calligraphy (Zhao Ji), was most accurately captured by the third option (generated by Uni-Calli)."

## I    MORE GENERATION RESULTS

This appendix provides a more extensive showcase of our model's capabilities in calligraphic style generation. It is divided into two main parts. The first part presents the complete poem "Bring in the Wine" (Qiang Jin Jiu) by Li Bai, generated in the distinct styles of several master calligraphers. This expands upon the teaser in the main text, where only a single line from each style was shown to form the poem collectively (see Figures 17, 18, and 16). The second part features a collection of names of famous ancient Chinese figures, rendered in various calligraphic styles (see Figures 15, 13, 14). This demonstrates our UniCalli's generalization ability and its proficiency in handling Chinese characters of varying complexity.

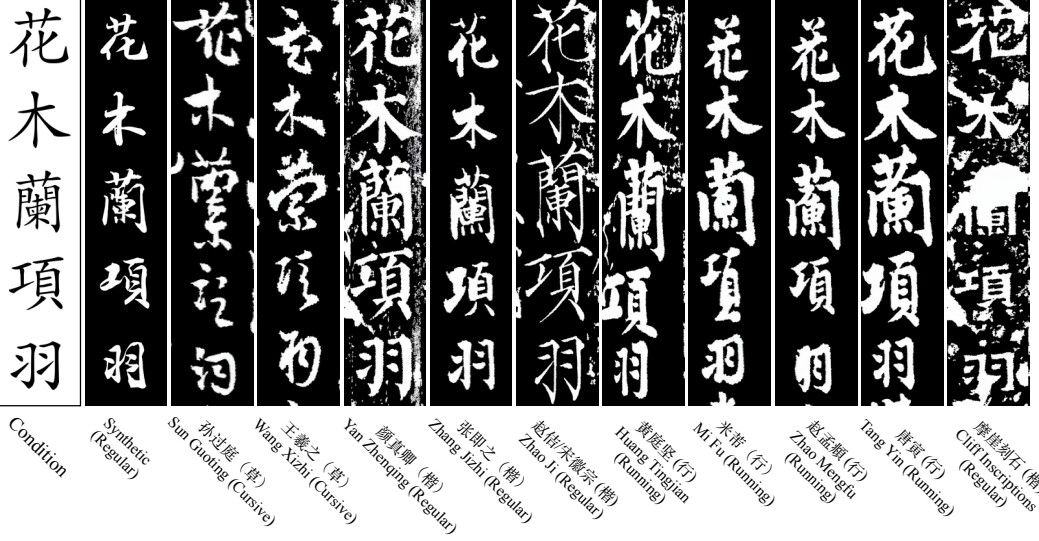

Figure 13: Names of historical and mythological figures from Chinese culture, rendered in various calligraphic styles. From top to bottom: Hua Mulan and Xiang Yu.

## J    APPLICATIONS IN CULTURAL AND CREATIVE INDUSTRIES

One of the primary motivations behind UniCalli is to bridge the gap between state-of-the-art generative AI and traditional cultural heritage. Beyond academic metrics, the true value of a calligraphy foundation model lies in its ability to empower designers and general users to create high-quality, aesthetically pleasing works for daily use with minimal technical barriers.

In this section, we showcase three specific types of cultural and creative products (Wen-Chuang) designed using UniCalli. It is important to highlight that the creation process for these works was remarkably streamlined. The calligraphic content was generated by our model, while the specific column layouts and ink color conversions were automated using simple post-processing scripts. The

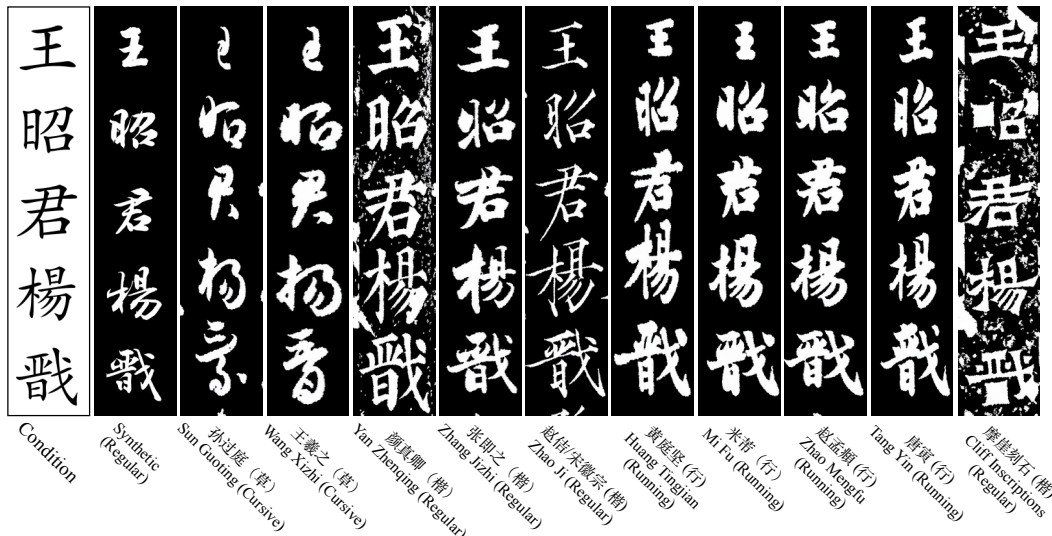

Figure 14: Names of historical and mythological figures from Chinese culture, rendered in various calligraphic styles. From top to bottom: Wang Zhaojun and Yang Jian.

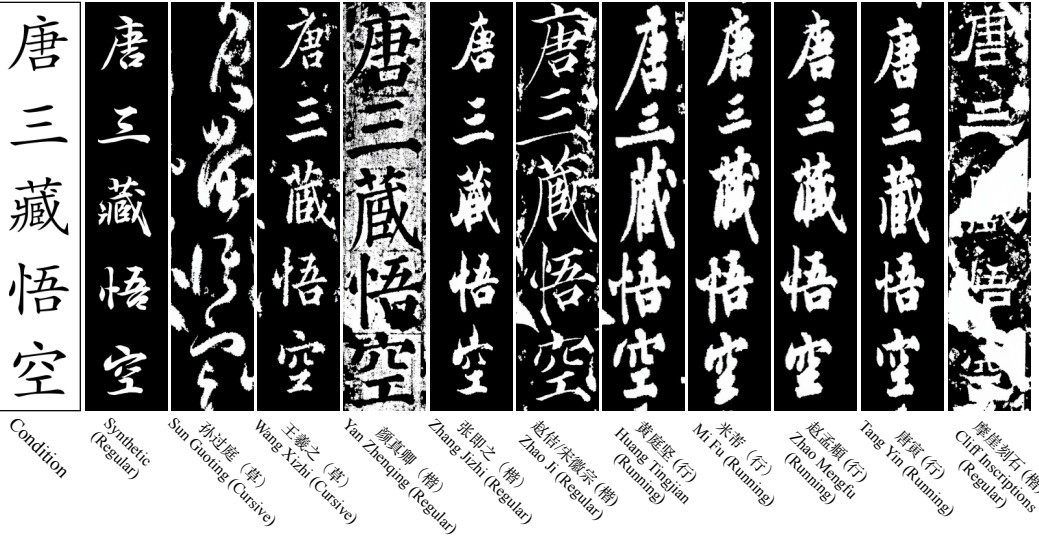

Figure 15: Names of historical and mythological figures from Chinese culture, rendered in various calligraphic styles. From top to bottom: Tang Sanzang and Wukong.

final compositions were produced by compositing these generated assets onto decorative background images sourced from the open internet. These examples demonstrate not only the model's capability to generate authentic, high-resolution calligraphy but also the ease with which it can be integrated into practical design workflows.

## J.1 CLASSICAL POETRY SCROLLS

The vertical column structure of Chinese calligraphy is most iconic in the presentation of classical poetry. We utilized UniCalli to generate complete scrolls of famous Tang and Song dynasty poems. The model maintains style consistency across multiple columns, capturing the flow (Qi-Yun) essential for artistic appreciation. Figure 19 presents a classical example featuring Meng Haoran's renowned poem "Spring Dawn," demonstrating the model's ability to generate extended vertical compositions with proper character spacing and rhythmic balance.

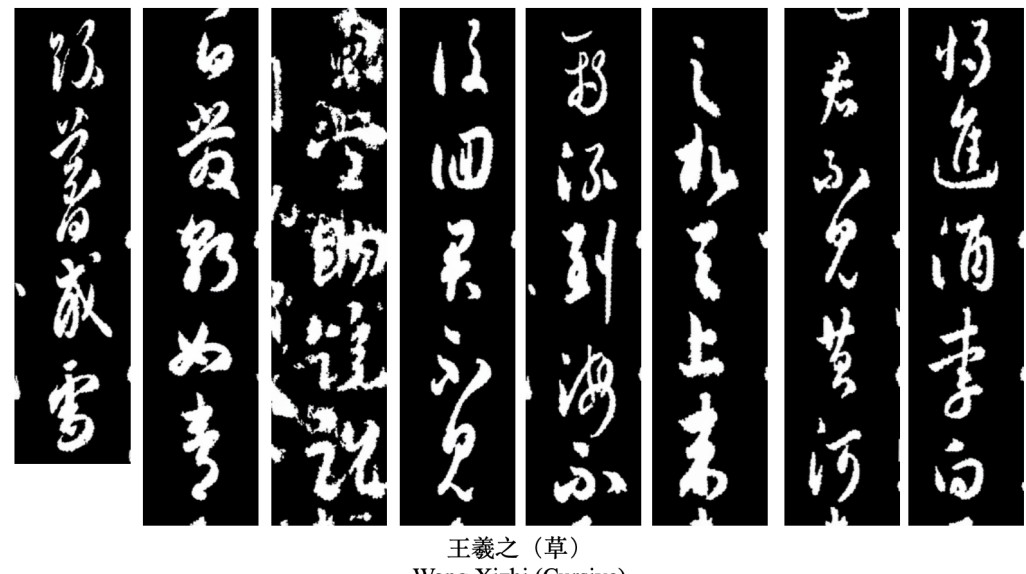

王羲之（草）
Wang Xizhi (Cursive)

Figure 16: The full text of Li Bai's "Bring in the Wine" (Qiang Jin Jiu), generated in the calligraphic style of Wang Xizhi (Cursive).

## J.2 BOOKMARKS AND AUSPICIOUS BLESSINGS

We also designed a series of cultural bookmarks and stationery featuring "Auspicious Phrases" (Ji-Li-Hua)—short, lucky idioms used to wish good fortune. Unlike fixed datasets, our text-to-image capability allows users to generate specific custom blessings in various artistic styles on demand, making them ideal for personalized gifts and souvenirs. Figures 20–23 showcase diverse calligraphic styles applied to popular blessing phrases such as "Good Luck," "Great Fortune," "Rolling Wealth," and "Pass Every Test." These examples feature the distinctive styles of master calligraphers including Mi Fu, Tang Yin, Zhao Ji, Zhao Mengfu, Sun Guoting, Wang Xizhi, and Huang Tingjian, demonstrating the model's versatility in style transfer while maintaining authentic character structures.

## J.3 TRADITIONAL MARRIAGE CERTIFICATES (HUN SHU)

In recent years, there has been a resurgence of interest in traditional Chinese marriage certificates (*Hun Shu*). These documents require a solemn, dignified, and elegant aesthetic. We applied UniCalli to generate customized marriage vows in multiple prestigious calligraphic styles. Figures 24–**??** present a collection of marriage certificates rendered in the styles of Zhao Ji, Zhao Mengfu, Mi Fu, Tang Yin, and Zhang Jizhi. The solemn and majestic character structures perfectly suit the formal nature of the occasion, while the model's ability to maintain consistency across multiple columns ensures the ceremonial dignity of the documents.

## J.4 SUMMARY OF IMPACT

These applications validate that UniCalli is not merely a style transfer tool but a production-ready asset generation engine. By open-sourcing this model, we hope to lower the barrier for designers to incorporate authentic Chinese calligraphy into modern cultural products, thereby fostering the revitalization of this intangible cultural heritage.

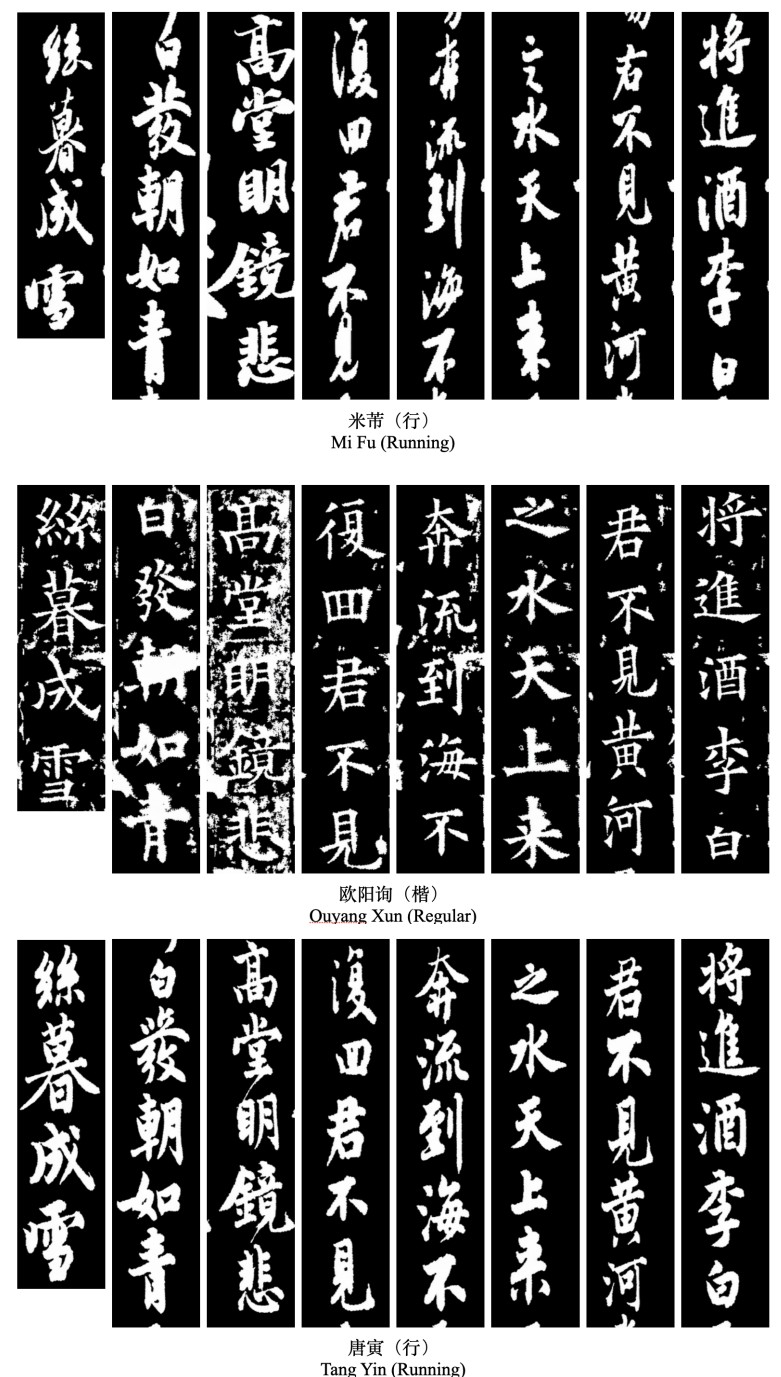

Figure 17: The full text of Li Bai's "Bring in the Wine" (Qiang Jin Jiu), generated in the calligraphic style of Mi Fu (Running), Ou Yangxun (Regular), and Tang Yin (Running).

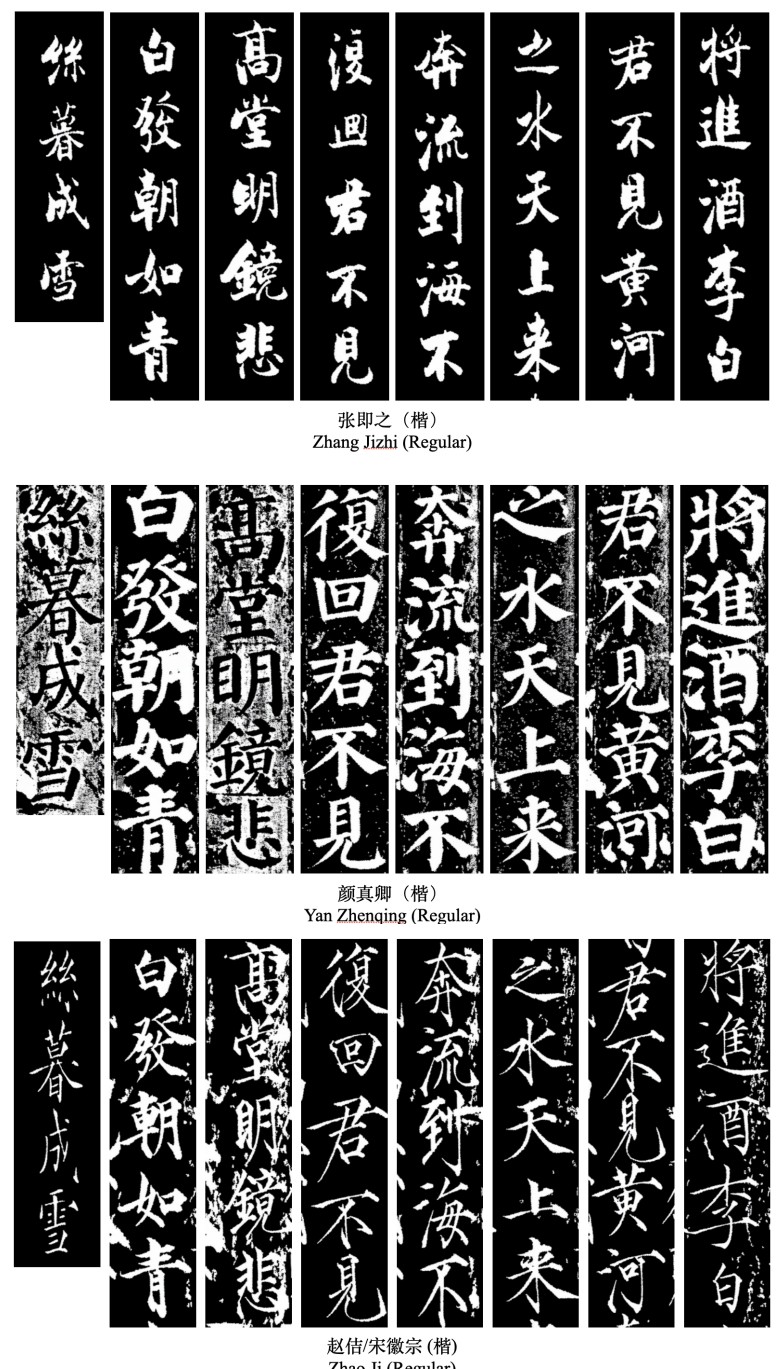

张即之（楷）
Zhang Jizhi (Regular)

颜真卿（楷）
Yan Zhenqing (Regular)

赵佶/宋徽宗 (楷)
Zhao Ji (Regular)

Figure 18: The full text of Li Bai's "Bring in the Wine" (Qiang Jin Jiu), generated in the calligraphic style of Zhang Jizhi (Regular), Yan Zhenqing (Regular), and Zhao Ji (Regular).

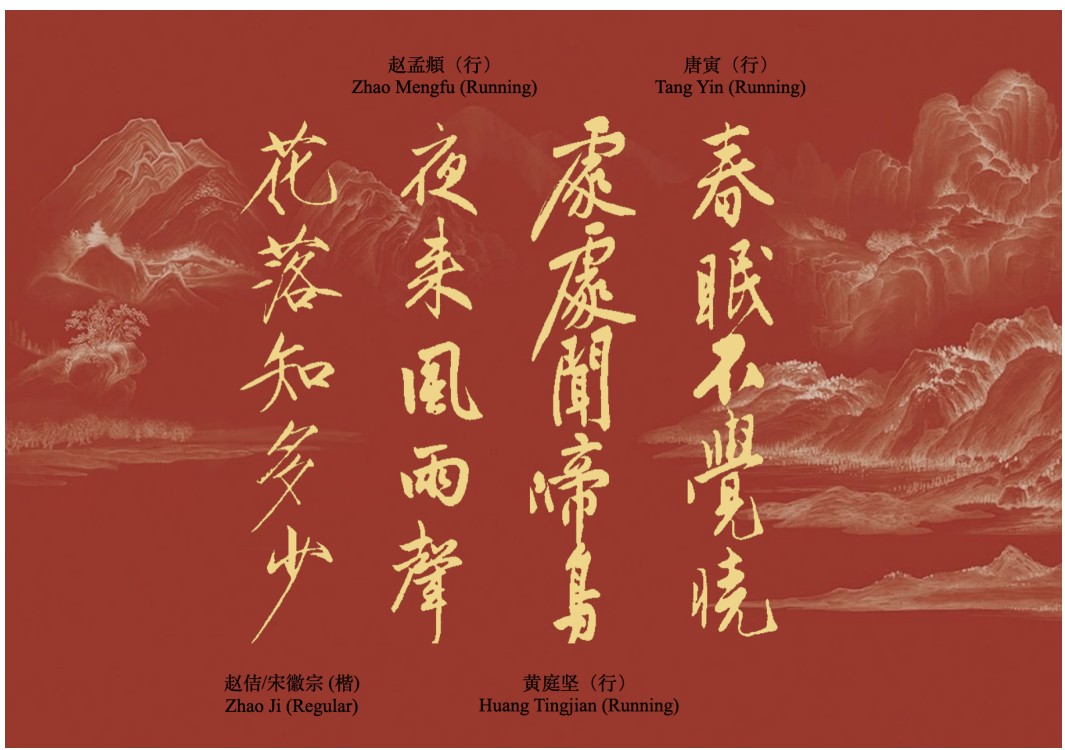

Figure 19: Meng Haoran - "Spring Dawn". Background images are sourced from the internet.

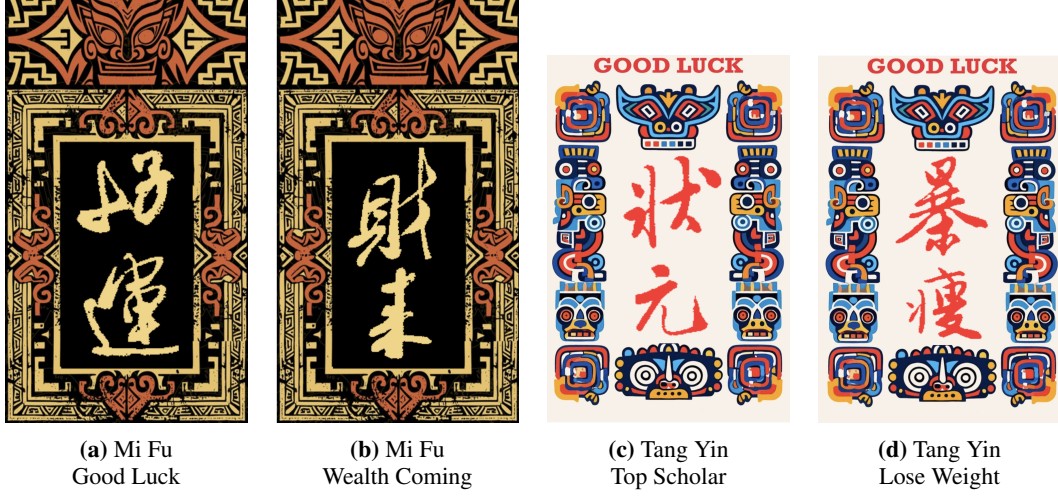

(a) Mi Fu
Good Luck

(b) Mi Fu
Wealth Coming

(c) Tang Yin
Top Scholar

(d) Tang Yin
Lose Weight

Figure 20: Cultural Creative Works - Group 1. Background images are sourced from the internet.

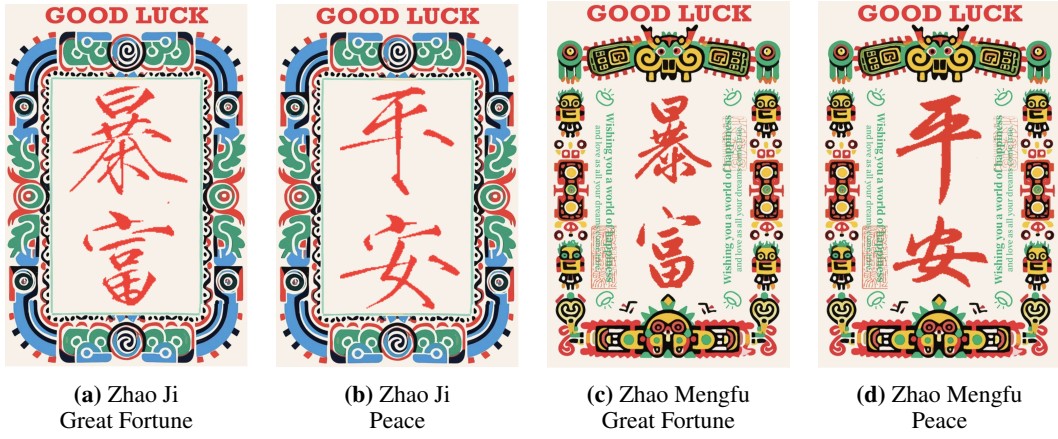

**(a)** Zhao Ji
Great Fortune

**(b)** Zhao Ji
Peace

**(c)** Zhao Mengfu
Great Fortune

**(d)** Zhao Mengfu
Peace

Figure 21: Cultural Creative Works - Group 2. Background images are sourced from the internet.

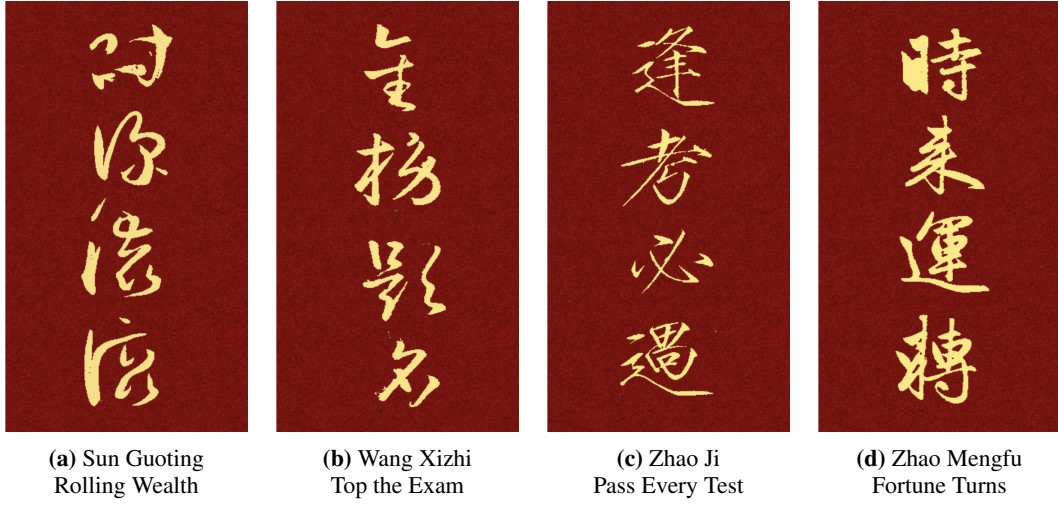

**(a)** Sun Guoting
Rolling Wealth

**(b)** Wang Xizhi
Top the Exam

**(c)** Zhao Ji
Pass Every Test

**(d)** Zhao Mengfu
Fortune Turns

Figure 22: Cultural Creative Works - Group 3. Background images are sourced from the internet.

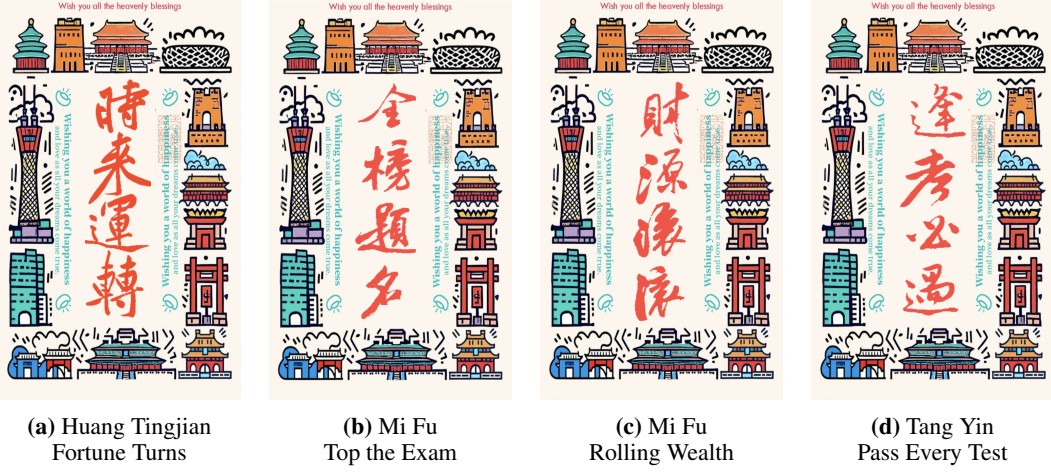

**(a)** Huang Tingjian
Fortune Turns

**(b)** Mi Fu
Top the Exam

**(c)** Mi Fu
Rolling Wealth

**(d)** Tang Yin
Pass Every Test

Figure 23: Cultural Creative Works - Group 4. Background images are sourced from the internet.

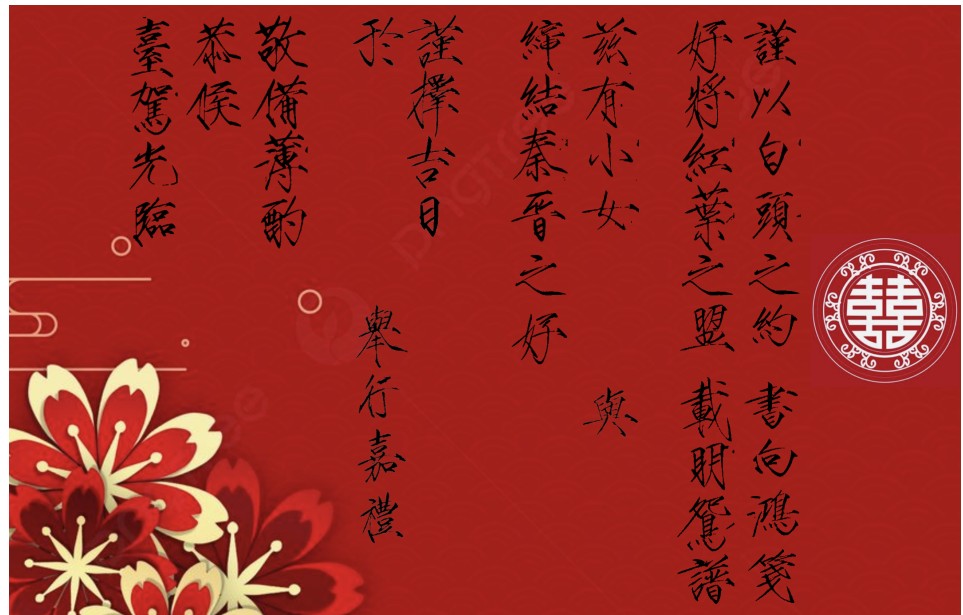

Figure 24: Zhao Ji - Wedding Certificate - Series 1. Background images are sourced from the internet.

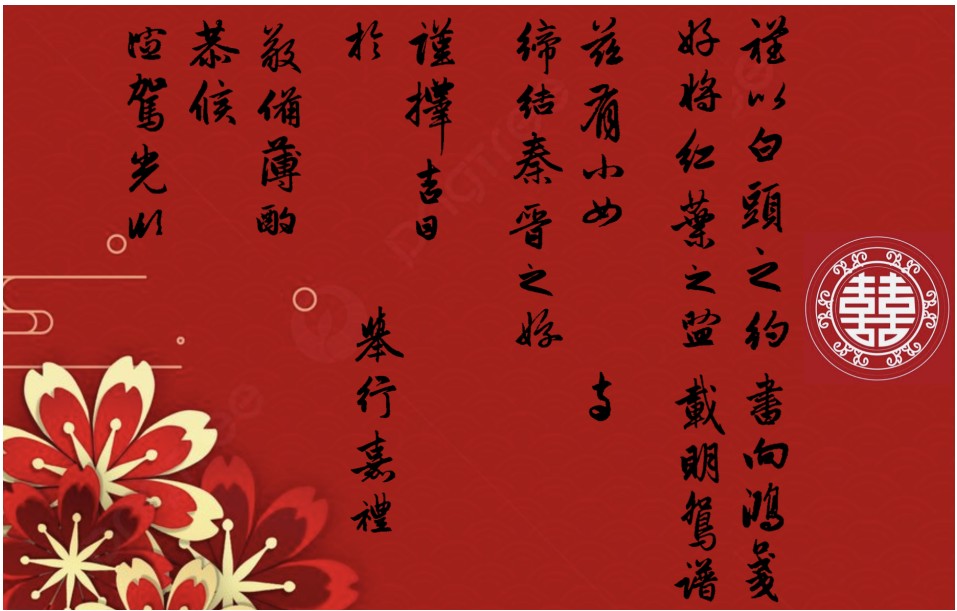

Figure 25: Zhao Mengfu - Wedding Certificate - Series 1. Background images are sourced from the internet.

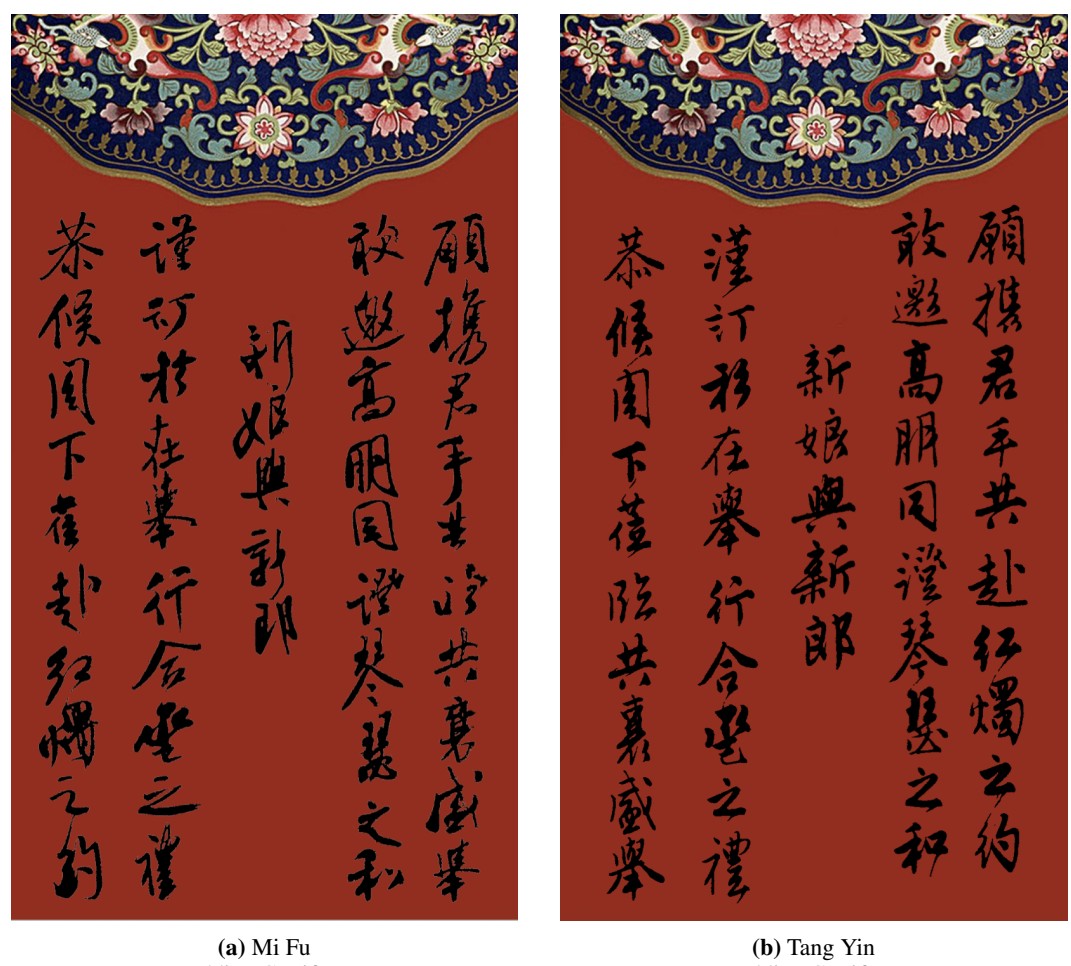

(a) Mi Fu
Wedding Certificate

(b) Tang Yin
Wedding Certificate

Figure 26: Wedding Certificates. Background images are sourced from the internet.

