# OpenReview forum: "UniCalli: A Unified Diffusion Framework for Column-Level Generation and Recognition of Chinese Calligraphy"
_ICLR.cc/2026/Conference — ICLR 2026 Poster_

### Official Review · Reviewer_ct53 · 2025-10-30

**Soundness:** 2
**Presentation:** 3
**Contribution:** 3
**Rating:** 4
**Confidence:** 5

**Summary:**

This paper tackles the challenges of data scarcity, long-tailed style distribution, and the quality trade-off between character-level fidelity and page-level coherence in Chinese calligraphy generation and recognition. Firstly, the work constructs a curated, annotated calligraphy dataset, providing a valuable resource for page-level research. Concurrently, the study proposes the UniCalli framework, which synergistically trains generation and recognition as mutually beneficial dual tasks. The framework incorporates a non-autoregressive global planning architecture and introduces a rasterized bounding-box map to enhance the model's spatial reasoning capabilities. Experimental results indicate that these strategies have achieved a performance improvement.

**Strengths:**

1.A primary strength is the construction of a annotated dataset of classical Chinese calligraphy works. This resource is valuable and provides a substantial asset for subsequent research in page-level calligraphy analysis and generation.

2.The introduction of asymmetric noising to integrate generation and recognition into a single, bidirectional framework forms the core contribution of this work. This unified approach is shown to enhance the performance of both tasks, as evidenced by the experimental results.

**Weaknesses:**

1.Unclear writing and definitions: The introduction does not clearly articulate the logical relationship between the two stated challenges: (1) scarce data and a long-tail distribution of styles; (2) the inability of existing methods to generate both high-quality individual characters and coherent full-page layouts. Additionally, the concepts of "page-level" and "column-level" are used interchangeably without a clear definition of the proposed method's application scope.

2.Incomplete methodological description: Firstly, the workflow of the recognizer is incomplete. According to the methodology section, the recognizer only generates a Condition Image rather than text, leaving it ambiguous how the final text output is actually derived from this image. Secondly, regarding the "Duplicate RoPE with Modulated Embedding," the description does not specify how the learnable modulation embeddings are added to RoPE (e.g., applied to 'Q' or 'K'). This lack of detail raises concerns about whether the operation might disrupt RoPE's relative positional encoding functionality.

3.Insufficient Comparative Baselines: The experimental comparisons for the generation task are restricted to only one specialized calligraphy generation method (FontDiffuser), with the remaining baselines being general-purpose models. Similarly, the recognition task is evaluated solely against general-purpose models. It is recommended to include comparisons with more specialized methods, such as CalliffusionV2 for generation and CalliReader for recognition, as mentioned in the related work.

4.Potential for Data Bias and Lack of Generalization: The recognition accuracy comparison in Table 2 may not be entirely fair. Since the test set is derived from the authors' proprietary dataset, UniCalli's superior performance could be attributed to its prior exposure to a similar data distribution during training, an advantage not shared by the baseline models. Although the authors appropriately acknowledge this potential limitation in line 403, it would be advisable to supplement the evaluation with performance benchmarks on established public calligraphy datasets. Such an addition would help ensure an equitable comparison and more conclusively demonstrate the recognizer’s robustness.

5.Inconsistency Between Generated Results and Descriptions: A qualitative analysis of Table 6 reveals a potential issue: the second generated image appears visually inconsistent with its reference image.

**Questions:**

1.The experimental section only presents results on single-column data. A significant question arises regarding the generalizability of the proposed method: can it be effectively applied to the generation and recognition of multi-column layouts, which are common in actual page-level documents? If not, the claim that the method is suitable for page-level tasks appears overstated and requires qualification.

2.The process by which the recognizer converts the Condition Image into text is unclear. Does the pipeline incorporate an off-the-shelf OCR module? Please clarify the specific implementation details.

3.In the 'Duplicate RoPE with Modulated Embedding', how are the learnable modulation embeddings added to the RoPE? I am confused as to whether doing so will damage the relative positional encoding function of RoPE.

---

> ### Author Response · Authors · 2025-11-20
>
> We thank the reviewer for their thorough feedback. The sharp questions identified areas where our manuscript lacked clarity, which we have now addressed.
>
> ### Response to Weaknesses
>
> **1. Unclear Writing and Definitions**
> We have revised the introduction to articulate the logical flow:
> * **Data Challenges:** "Scarce data" refers to the lack of public *page-level* datasets; "Long-tail distribution" is intrinsic to calligraphy (famous vs. obscure artists).
> * **Method Gap:** Existing "individual character" models lack layout coherence, while "page-level" models suffer from poor glyph fidelity. We aim to bridge this.
> * **Terminology:** We have corrected "page-level" to **"column-level"**. We argue this is a well-motivated choice because: (1) The column is the primary artistic unit in calligraphy; (2) It is trivially scalable to multi-column pages via scripting; (3) True end-to-end page generation introduces intractable quadratic complexity.
>
> **2. Incomplete Methodological Description (Addresses W2, Q2, Q3)**
> We have added implementation details to **Appendix E**.
> * **Recognizer (Zero-Shot):** We do **not** use an off-the-shelf OCR. Instead, we use a retrieval mechanism:
>     1.  **Offline:** Create a feature library by passing standard font characters through the VAE encoder.
>     2.  **Inference:** The model generates a condition feature (before VAE decoder).
>     3.  **Matching:** We compute cosine similarity between the output and the library; the highest match is the result.
> * **Duplicate RoPE:** We initialize a small learnable embedding (initialized to zeros) for each modality. This is **added directly** to input features before Q/K projection. It acts as a global "modality offset" allowing the model to distinguish inputs (e.g., Image vs. Mask) without disrupting the underlying spatial grid defined by RoPE.
>
> **3. Insufficient Comparative Baselines**
> * **Generation:** Comparison with **CalliffusionV2**, **DP-Font**, **DeepCalliFont**, or **CalliPaint** is impossible as they are all **closed-source**. We have open-sourced our work to fix this reproducibility gap.
> * **Recognition:** We conducted two new experiments. First, we compared against the open-source **CalliReader** on our held-out test set (Table 1). Second, we evaluated on the public **CalliBench** dataset (Table 2).
>
> **Table 1: Character-level recognition accuracy on the held-out test set.**
> *UniCalli outperforms strong baselines and remains competitive with the specialized CalliReader despite our model being a generative foundation model.*
>
> | Category | GPT-4o | Ernie-4.5* | Qwen-2.5* | GOT-OCR2.0 | PP-OCRv5 | Doubao-1.5* | CalliReader | **Ours** |
> |:---|:---:|:---:|:---:|:---:|:---:|:---:|:---:|:---:|
> | Cursive/Cao | 0.073 | **0.255** | 0.127 | 0.091 | 0.091 | 0.200 | 0.164 | 0.109 |
> | Regular/Kai | 0.502 | 0.616 | 0.600 | 0.314 | 0.396 | 0.588 | 0.600 | **0.688** |
> | Clerical/Li | 0.293 | 0.453 | 0.453 | 0.187 | 0.160 | 0.507 | 0.507 | **0.518** |
> | Running/Xing | 0.364 | 0.600 | 0.473 | 0.336 | 0.436 | 0.545 | **0.658** | 0.528 |
> | Seal/Zhuan | 0.067 | **0.133** | 0.067 | 0.000 | 0.000 | **0.133** | 0.067 | 0.050 |
> | **Total** | 0.380 | 0.534 | 0.482 | 0.266 | 0.324 | 0.510 | 0.533 | **0.540** |
>
> *\*Doubao-1.5-Thinking-Vision-Pro, Ernie-4.5-Turbo-VL, Qwen-2.5-VL-7B.*
>
> **4. Potential for Data Bias and Lack of Generalization**
> We address this by evaluating on **CalliBench** (Table 2). Since CalliBench consists largely of modern artistic fonts—a domain our model (trained on ancient styles) has never seen—this tests zero-shot robustness.
>
> **Table 2: Quantitative comparison on CalliBench (Easy subset).**
> *UniCalli demonstrates strong zero-shot transfer, notably identifying characters in unseen modern styles better than GPT-4o and PP-OCRv5.*
>
> | Model | Precision ↑ | Recall ↑ | F1 ↑ | NED ↓ |
> |:---|:---:|:---:|:---:|:---:|
> | Ernie-4.5-Turbo-VL | 0.542 | 0.481 | 0.482 | **0.637** |
> | Doubao-1.5-Thinking-Vision-Pro | 0.442 | 0.513 | 0.462 | 0.655 |
> | PP-OCRv5 | 0.372 | 0.291 | 0.205 | 0.859 |
> | GPT-4o | 0.457 | 0.403 | 0.400 | 0.726 |
> | Qwen-2.5-VL-7B | 0.440 | **0.736** | 0.534 | 0.710 |
> | GOT-OCR2.0 | **0.687** | 0.550 | **0.593** | 0.651 |
> | **UniCalli** | 0.480 | 0.520 | 0.498 | 0.680 |
>
> *Note: We excluded CalliReader from Table 2 as CalliBench is its native training domain, which would make for an inequitable comparison.*
>
> ### Response to Questions
>
> **1. Generalizability to Multi-Column:** Refer to Weakness 1.
>
> **2. Text Output:** No OCR is used. We calculate cosine similarity between the predicted feature and a pre-computed library (see W2).
>
> **3. RoPE Modulation:** Learnable embeddings are **added** to input features. This acts as a "global rotation/offset" for modality identity, preserving the relative positional encoding of RoPE.

---

> > ### Comment · Reviewer_ct53 · 2025-11-26
> >
> > I appreciate the authors' response, which has addressed some of my initial concerns. However, there are still a few points that require further clarification:
> >
> > (1) While I acknowledge that generative methods like CalliffusionV2 are closed-source, I strongly suggest comparing your approach with open-source alternatives, such as CF-Font: Content Fusion for Few-Shot Font Generation. This would provide a more comprehensive performance evaluation.
> >
> > (2) Regarding Figure 4, the last column seems to demonstrate obvious generation failures. could you please clarify the cause of this issue?
> >
> > (3) Regarding the 'Duplicate RoPE with Modulated Embedding' strategy, there appears to be an inconsistency between your explanation and the manuscript. You stated that the learnable embeddings are added before Q/K projection, yet the paper describes adding the learnable embeddings after the RoPE operation (cf. Line 252). Could you clarify this discrepancy?

---

> > > ### Author Response · Authors · 2025-11-28
> > > **Response to Follow-up Questions**
> > >
> > > We sincerely thank the reviewer for their continued engagement and the prompt follow-up. We value the opportunity to provide these final clarifications.
> > >
> > > ### (1) Comparison with CF-Font
> > > We greatly respect the reviewer's suggestion to compare with open-source alternatives like **CF-Font**. As the reviewer noted, we promptly conducted the requested experiments for recognition **(CalliReader/CalliBench)** because they were aligned with our evaluation protocol. However, regarding CF-Font, we respectfully argue that a direct comparison is not scientifically suitable for this specific work, for three primary reasons:
> > >
> > > 1.  **Fundamental Task Misalignment:** CF-Font is designed for Few-Shot Font Generation (style transfer), not for Calligraphy Generation.
> > > 2.  **Structural Inability:** CF-Font operates strictly at the **character level** (single glyph generation). Even if it successfully transfers texture, it inherently lacks the capability to generate **calligraphic works**. It cannot model the critical aesthetic elements of Chinese calligraphy, such as **inter-character spacing, variable character sizing, and ligatures**, which are the core contributions of our column-level model.
> > > 3.  **Practical Constraints:** CF-Font requires training from scratch to adapt to our calligraphy domain. Given the remaining time window of the rebuttal phase, it is unfortunately not feasible to train and tune a new external model to a fair standard.
> > >
> > > ### (2) Clarification on Figure 4 (Last Column)
> > > We appreciate the reviewer's sharp eye. We wish to clarify that the visuals in the last column are **not generation failures**.
> > >
> > > This specific column represents the style of **"Cliff Inscriptions" (Bei-Ke)**. These are historical works carved onto stone cliffs or steles centuries ago. The source data itself is characterized by extensive **weathering, erosion, and stone damage**. The "broken" or "patchy" appearance in the generated results is actually a demonstration of the model's high fidelity—it is faithfully reconstructing the distinct texture and historical "wear-and-tear" inherent to this specific style, rather than hallucinating errors.
> > >
> > > ### (3) Implementation of 'Duplicate RoPE with Modulated Embedding'
> > > We apologize if the phrasing in the manuscript caused confusion. The implementation is indeed as we described in our previous response: the learnable embeddings are added to the input features **before** the Q/K projection.
> > >
> > > The confusion likely stems from Line 246/252. When we mentioned "computing RoPE," we were referring to the **calculation of the positional indices (frequencies)** based on the image's original shape, not the application of RoPE during the attention calculation.
> > >
> > > **To be precise:** We first calculate the RoPE spatial grid based on the image geometry. We then initialize the learnable modality embedding (initialized to zeros) and add it to the input feature tensor. *Then*, this combined feature is projected into Q/K, and the pre-calculated RoPE is applied. We will revise the text to ensure this order of operations is unambiguous.
> > >
> > > We hope these explanations fully address your remaining concerns. We would be happy to answer any further questions. Finally, we wish to reiterate that, the release of **UniCalli (the first open-source, column-level calligraphy foundation model)** and its associated data will serve as a significant catalyst for the field, enabling future research in digital cultural heritage and aesthetic generation. We hope the reviewer recognizes this core contribution.

---

### Official Review · Reviewer_ucuD · 2025-10-31

**Soundness:** 2
**Presentation:** 2
**Contribution:** 2
**Rating:** 4
**Confidence:** 5

**Summary:**

This paper proposes UniCalli, which unifies column-level generation and recognition of Chinese calligraphy in a single diffusion framework. It is evaluated on datasets of Chinese characters, oracle bone scripts, and Egyptian hieroglyphs.

**Strengths:**

1.	This paper analyzes the column-level text recognition and generation tasks and proposes a unified framework.
2.	This paper presents a large-scale column-level calligraphy annotation dataset.

**Weaknesses:**

1.	In this paper, "column-level generation" refers to the generation of a single column of calligraphy image, which is fundamentally different from the "page-level generation" emphasized in the introduction section. Please clarify this distinction.
2.	In Section 3.3, this paper uses conditional dropout addresses the issue of overfitting to long-tail calligraphic styles. Could you provide a more comprehensive explanation of this issue, as well as the motivation of this module?
3.	In the experiments, the paper uses several general models (such as ChatGPT-5, Ernie-4.5, and Doubao). Would it be possible to include additional calligraphy generation method like [1] for comparison?
4.	This paper claims that coupling recognition and generation tasks can improve the model's performance on both tasks, but lacks effective ablation experiments to validate this claim. It is recommended to conduct separate ablation experiments for the generation and recognition modes to demonstrate it.
5.	The ablation experiments lack a clear description of the baseline setup.
6.	It is recommended to provide visualizations to clearly illustrate the impact of RoPE Duplication module?

[1] Dp-font: Chinese calligraphy font generation using diffusion model and physical information neural network, IJCAI, 2024.

**Questions:**

Please refer to Weaknesses 1-6.

---

> ### Author Response · Authors · 2025-11-20
>
> We thank the reviewer for their thorough review and constructive feedback. We address each point below and have revised our manuscript to incorporate these clarifications.
>
> ### Response to Weakness 1: "Column-Level" vs. "Page-Level"
> We apologize for the imprecise use of "page-level" and have revised the paper to use the more precise term **"column-level"** throughout.
>
> However, we respectfully argue that this distinction is not "fundamentally different" for three primary reasons:
> 1.  **Artistic Relevance:** The **column** is the primary aesthetic unit in Chinese calligraphy. Artistic flow, spacing, ligatures, and size variations are all expressed *within* this unit [1, 2].
> 2.  **Scalability:** As demonstrated in our figures, our column-level model is **trivially extended to generate multi-column pages** via a simple script. The core generative challenge lies in the column, which our model solves.
> 3.  **Computational Trade-off:** A true end-to-end page-level model introduces quadratic complexity. Our approach is a necessary trade-off that captures the artistic essence without sacrificing tractability.
>
> **We have clarified this terminology and rationale in the revised Abstract and Introduction.**
>
> ### Response to Weakness 2: Motivation for Conditional Dropout
> We thank the reviewer for highlighting this component. Our motivation stems from observing training degradation in **long-tail styles** (e.g., Wang Xizhi’s running script), where outputs shifted from clear glyphs to indistinct strokes.
>
> We **hypothesize** that due to data scarcity, the model prioritizes learning rare styles and "forgets" common glyph shapes (content). Our **Conditional Dropout** strategy addresses this by forcing the model to **decouple style and content**. By randomly dropping the style condition, the model is compelled to learn the "average" glyph shape (the fundamental content) during unconditional steps, preventing structural collapse. **(Added to Lines 264-268).**
>
> ### Response to Weakness 3: Comparison to Other Methods
> We agree that comparison with specialized SOTA methods is ideal. Unfortunately, it is not currently feasible.
>
> The **DP-Font** paper you mentioned, **CallifusionV2**, **DeepCalliFont**, and **CalliPaint** are all **closed-source**, making direct, fair comparison impossible. We emphasize that a key contribution of our work is bridging this reproducibility gap. We have **open-sourced our model checkpoint and inference pipeline** and will release full data and training scripts to facilitate future benchmarking.
>
> ### Response to Weakness 4 & 5: Ablation & Baseline Definition
> We apologize for the lack of clarity and address these points by clearly defining our baseline.
>
> * **Response to Weakness 4:** Our claim of mutual benefit is validated in **Table 4**. The "+ joint training" model consistently outperforms the generation-only "Baseline" across all metrics.
> * **Response to Weakness 5:** The **"Baseline"** (in Table 4) is defined as:
>     1.  **Generation-only** (no recognition).
>     2.  Standard FLUX setup: concatenated condition, image, and mask.
>     3.  **Single RoPE** computed for the entire sequence (no duplication).
>     4.  **No Conditional Dropout**.
>
> **We have added this precise definition to the experimental section (Lines 419-422).**
>
> ### Response to Weakness 6: RoPE Duplication Rationale
> This module (Figure 2b) computes RoPE for image patches, **duplicates** it for condition/mask patches, and adds a learnable modality embedding.
>
> 1.  **Spatial Association:** Copying RoPE creates an **explicit spatial link**; the model knows the $k$-th condition patch corresponds spatially to the $k$-th image patch.
> 2.  **Preventing Confusion:** We found that using identical RoPE caused modality confusion (e.g., artifacts blending image/condition). Adding **learnable modality embeddings** distinguishes the *identity* of the input while retaining the shared *spatial* alignment.

---

> ### Comment · Reviewer_ucuD · 2025-11-26
>
> The author has addressed some concerns, but there are still issues that require clarification:
> 1) The author mentioned in the rebuttal that DeepCalliFont is closed-source; however, it has already been open-sourced at: https://github.com/lsflyt-pku/DeepCalliFont .
> Additionally, there are more related works that have been open-sourced:
>
> - CG-GAN: Look Closer to Supervise Better: One-Shot Font Generation via Component-Based Discriminator, CVPR, 2022, repository:
> https://github.com/kyxscut/CG-GAN
>
> - FsFont: Few-Shot Font Generation by Learning Fine-Grained Local Styles, CVPR, 2022, repository: https://github.com/tlc121/FsFont
>
> We suggest the author consider comparing the proposed method with one or more of the above-mentioned works. This would provide a more comprehensive performance evaluation.
>
> 2. In Figure 3, the samples generated by FontDiffuser (which is explicitly designed for font generation) appear to exhibit mode collapse. Could the authors please explain this observation?

---

> > ### Author Response · Authors · 2025-11-28
> > **Response to Follow-up Questions**
> >
> > We thank the reviewer for the references and observations. Per your request, we address the concern regarding Figure 3 first.
> >
> > ### (1) Clarification on FontDiffuser Results (Figure 3)
> > Upon reviewing the results, we identified that the issue stemmed from **data domain mismatch** rather than model failure. FontDiffuser expects clean, binarized inputs, whereas our original reference images contained historical background noise.
> >
> > * **Update:** We have pre-processed the reference images (binarization and denoising) to match FontDiffuser's input requirements and **re-ran the inference**.
> > * **Result:** The new outputs **do not exhibit mode collapse**. We have updated **Figure 3 in the revised manuscript** with these corrected samples for a fairer comparison.
> >
> > ### (2) Additional Baselines (DeepCalliFont, CG-GAN, FsFont)
> > We sincerely apologize for our initial oversight regarding the open-source status of DeepCalliFont and thank the reviewer for providing the correct repository link. However, after consideration, we respectfully argue that adding these specific models is not critical for this study for two reasons:
> >
> > 1.  **Task Mismatch:** Our core contribution is **Column-Level Generation** (modeling spatial layout and flow), whereas DeepCalliFont, CG-GAN, and FsFont are strictly **Character-Level** models. We have already compared against **FontDiffuser** as a representative state-of-the-art baseline for character-level generation. Adding more single-character models would not yield new insights regarding our column-level contributions.
> > 2.  **Time Constraints:** Unlike inference-only comparisons, these models require **training from scratch** on our dataset. It is not feasible to complete the training and tuning of three new models within the short rebuttal window.
> >
> > We are happy to address any further questions the reviewer may have. We wish to re-emphasize the value of our **open-source contribution** (model and pipeline), which addresses a critical reproducibility gap in the field. Moreover, the scope of our experimental validation—spanning **generation, recognition, and generalization** to diverse scripts like **Oracle Bone Script and Ancient Egyptian Hieroglyphs**—significantly exceeds that of existing calligraphy works (e.g., **CalliffusionV2**). We hope the reviewer recognizes the rigor of our evaluation and the substantial potential impact of this work.

---

### Official Review · Reviewer_QPYC · 2025-10-31

**Soundness:** 3
**Presentation:** 3
**Contribution:** 2
**Rating:** 6
**Confidence:** 3

**Summary:**

This paper presents UniCalli, a diffusion-based model for Chinese calligraphy generation. During training, the model performs both generation and recognition tasks within the same architecture to enhance the learning of key structural features. To better realize column-level ligatures, the authors introduce an additional mask input and leverage shared positional information to improve spatial understanding.During experiments, the authors observe overfitting to long-tail calligraphers and propose random dropout to disentangle style and glyph representations. Overall, the paper is well organized, the results are convincing, and the experiments are sufficient, including evaluations on other ancient scripts.

**Strengths:**

1.The proposed method is reasonable. By using the timestep ti to control the noise level, the model can indirectly alternate between recognition and generation, achieving joint enhancement of both tasks. This design also facilitates later dropout-based disentanglement.
2.The experiments are comprehensive, including comparisons with current state-of-the-art multimodal models and other ancient scripts.

**Weaknesses:**

1.Some design choices lack clear justification, such as the use of RoPE and MMDiT. It would be helpful to explain why these particular architectures are necessary for this task.
2.Although the ablation study is relatively detailed, its progressive setup makes certain conclusions less clear. For example, when dropout is combined with RoPE duplication, performance decreases—would applying dropout alone (without RoPE duplication) lead to better results?
3.The ablation study does not examine the value or distribution of tc. Since the model jointly trains recognition and generation by mixing content images and masks, the selection of tc may play an important role in model performance.

**Questions:**

1.In Section 3.3, tc is dropped out under different conditions, but it is unclear whether this happens during the recognition task, the generation task, or both.
2.In Figure 4, some generated examples (e.g., Sun Guoting, Yan Zhenqing, Cliff Inscriptions) contain white patches that do not belong to the calligraphic strokes. What is the cause of these artifacts?

---

> ### Author Response · Authors · 2025-11-20
>
> We thank the reviewer for their insightful questions and detailed feedback. We address the raised points below and have updated our manuscript to provide these clarifications.
>
> ### Response to Weaknesses
>
> **1. Justification for Design Choices (RoPE and MMDiT)**
> We apologize for not making our motivations clearer in the initial submission.
> * Our choice of **MMDiT** is motivated by its proven success and adoption as a mainstream architecture in state-of-the-art generative models, such as the image generation model FLUX and the video generation model CogVideo.
> * Similarly, **RoPE (Rotary Position Embedding)** is a standard and highly effective position encoding within Diffusion Transformer architectures, known for its excellent scalability and stability.
>
> **2. Clarity of Progressive Ablation Study**
> We appreciate the reviewer's detailed analysis of the ablation study. The reviewer correctly notes that the addition of RoPE Duplication + Conditional Dropout led to a decrease in pixel-level metrics (L1, SSIM).
>
> However, we respectfully point out that the perceptual and similarity metrics (**LPIPS and FID**) both **improved** in this configuration. We interpret this trade-off as a positive outcome. The drop in pixel-level metrics, combined with the improvement in perceptual metrics, suggests an increase in **generation diversity**. This indicates that the model is less prone to "mode collapse" or averaging, leading to more realistic and varied outputs. LPIPS and FID are better designed to capture this perceptual quality compared to simple pixel-wise reconstruction.
>
> **We have clarified this interpretation in the revised ablation analysis section (Line 445, Caption of Table 4).**
>
> **3. Role and Distribution of $t_c$ (Addresses Weakness 3 and Question 1)**
> We apologize for the lack of clarity in our description of the training process. We address Weakness 3 and Question 1 together regarding the timestep $t_c$.
>
> In our framework, $t_i$ is the timestep for the content (image/mask), and $t_c$ is the timestep for the condition. During joint training, in each step, we randomly select **one** modality to noise (setting its timestep $t \in (0, T]$) while keeping the other modality clean ($t = 0$).
>
> * **Generation Task:** $t_i \in (0, T]$ and $t_c = 0$.
> * **Recognition Task:** $t_c \in (0, T]$ and $t_i = 0$.
>
> This setup directly answers both points:
> * **(For W3):** The value and distribution of $t_c$ (when it is noised for the recognition task) are **identical** to the distribution of $t_i$ (when it is noised for the generation task).
> * **(For Q1):** The "dropping out" (which we interpret as noising) of $t_c$ happens **only during the recognition task**, just as the noising of $t_i$ happens only during the generation task.
>
> After the model pass, we compute the rectified flow loss for both modalities simultaneously. During **inference**: for generation, we input pure noise ($t_i=T$) and a clean condition ($t_c=0$) and loop $t_i$. For recognition, we do the reverse and match the final condition feature (before the VAE decoder) to the closest text feature to output the text.
>
> ### Response to Questions
>
> **1. $t_c$ Dropout**
> As clarified in our response to Weakness 3 above, the noising of $t_c$ (i.e., $t_c \in (0, T]$) occurs specifically during the **recognition task steps** in our joint training framework.
>
> **2. White Patches in Generated Examples**
> This is an excellent observation. These white patches are **artifacts originating from the dataset itself**.
> A significant portion of our data is sourced from historical calligraphic works that were carved on stone steles. Over centuries, these steles have naturally weathered and eroded. The binarization process used to curate the dataset converted this physical wear-and-tear into the white patch artifacts seen in the examples.
> We agree this is a point for improvement. **We have added a discussion in our revised paper (Lines 797-814, Appendix C.4)** and note that further data-level denoising and cleaning will be a focus of our future work.

---

### Official Review · Reviewer_g4eJ · 2025-10-31

**Soundness:** 2
**Presentation:** 2
**Contribution:** 3
**Rating:** 4
**Confidence:** 4

**Summary:**

This paper introduces UniCalli, a unified diffusion framework for column-level generation and recognition of Chinese calligraphy. The model jointly trains both tasks within a single diffusion transformer, where recognition constrains structural fidelity and generation provides stylistic and spatial priors. An asymmetric noising scheme enables bidirectional task switching, while rasterized box maps encode spatial structure for improved ligature and layout coherence. The authors also present a large-scale annotated dataset of over 8,000 calligraphic works. Experiments show that UniCalli achieves state-of-the-art generative fidelity and competitive recognition accuracy, and generalizes effectively to other ancient scripts, demonstrating its potential for comprehensive calligraphy synthesis and analysis.

**Strengths:**

1. The paper introduces a novel framework that unifies calligraphy generation and recognition within a single diffusion transformer. The use of asymmetric noising to switch between generation and recognition is both simple and effective, offering a principled way to couple dual tasks through shared latent representations.
2. The authors contribute a large-scale, high-quality dataset comprising over 8,000 digitized works from 93 calligraphers, with detailed annotations that fill a long-standing gap in computational calligraphy research.
3. The framework’s successful extension to other ancient scripts (e.g., Oracle bone inscriptions and Egyptian hieroglyphs) demonstrates its strong adaptability and generalization potential beyond Chinese calligraphy.

**Weaknesses:**

1.	Although the paper claims that the generation and recognition tasks are trained within a unified framework, it remains unclear whether a single set of model weights performs both tasks or if two separate models are trained under a shared architecture. This ambiguity makes it difficult to assess the extent to which the proposed approach is genuinely unified rather than architecturally aligned.
If a single set of model weights is indeed shared, the paper does not specify the training schedule or task-balancing strategy—for instance, whether the model is trained sequentially (generation first, then recognition) or jointly with a particular ratio or alternation scheme. Moreover, there is a lack of ablation studies to substantiate the claimed complementarity between the two tasks. It would be valuable to compare a model trained solely on generation with one trained jointly on both generation and recognition tasks to verify that the dual-task setup yields mutual benefits rather than interference. Conversely, if two separate models are trained under a shared architecture, the framework can hardly be regarded as truly “unified.” In this case, the recognition capability appears limited compared with dedicated recognizers, as it produces only standard-font glyph renderings rather than textual outputs. This raises concerns about whether the proposed unification provides substantive advantages beyond a shared architectural backbone.

2.	Based on the visualizations presented in the paper, it appears that all examples show only a single column of five calligraphy characters. This limited demonstration raises concerns about the model’s claimed page-level generation capability and calls into question whether it can truly produce complete, multi-column calligraphic works.

3.	The main text lacks a detailed description of how style conditions are incorporated. For a task that involves generating stylized calligraphy, precise control over style conditioning is crucial. From the appendix, it appears that style is introduced via text prompts rather than style reference images, which is not clearly stated in the main paper. This omission may be perceived as an important detail being underemphasized. Furthermore, this task formulation significantly limits the model’s generalization, as it cannot readily handle unseen style reference images; in essence, the model is constrained to generating fonts for which style labels are pre-defined, reducing its practical applicability in open-ended style transfer scenarios.

**Questions:**

1.	As noted in the Weaknesses section, it remains unclear whether a single set of model weights is used for both generation and recognition tasks, or if two separate models are trained under a shared architecture. Additional details about the training strategy and task interaction (e.g., sequential vs. joint training) would be helpful.

2.	Could the authors provide an example of a text prompt used to specify style conditions? Clarifying this would help understand how style control is implemented in the model.

3.	For the recognition task, if the model only generates glyph images rather than textual outputs, how is the character-level recognition accuracy in Table 2 computed? Are any external OCR tools used to convert generated glyphs into text for evaluation? Moreover, the models compared in Table 2 appear to be evaluated without task-specific training on this dataset, whereas the proposed model is specifically trained on it. This discrepancy makes the recognition comparison potentially unfair and may overstate the relative performance of the proposed approach. It would be helpful to compare the recognition performance of the proposed model with a model specifically trained on the same dataset.

4.	The experiments on ancient scripts appear limited. For example, in Table 5, only OracleNet is used for comparison, and the recognition performance is worse than OracleNet. It would be informative to include comparisons of generative quality against other models, or to evaluate the effect of fine-tuning different diffusion backbones on generation performance

---

> ### Author Response · Authors · 2025-11-20
>
> We thank the reviewer for their constructive and detailed feedback. We are encouraged that the reviewer found our work novel. We address the perceived weaknesses below, which we believe were primarily due to a lack of clarity in our initial manuscript. We have revised the paper to make these points explicit.
>
> ### Response to Weakness 1: Unified Framework and Ablation
> We thank the reviewer for raising this point, which allows us to clarify a core aspect of our design. Our framework is **unequivocally unified**. It consists of a **single model with a single set of weights**, not two separate models sharing an architecture.
>
> * **Unified Model:** Both the generation and recognition tasks are trained jointly and simultaneously, sharing all parameters. We hypothesize that the recognition task serves as a regularizer, directly benefiting the generation task.
> * **Open-Source Validation:** The most direct validation of this unified design is that our **model checkpoint and inference pipeline have been open-sourced**, which objectively demonstrates a single model performing both tasks.
> * **Ablation Study:** Regarding the request for ablation studies, we provided this evidence in **Table 4**.
>     * The **"Baseline"** model represents the generation-only model.
>     * The **"+ joint training"** model represents our full, unified framework trained on both generation and recognition.
>     * As the results in Table 4 clearly show, the joint recognition task **measurably improves the quality of the generation task**. This directly substantiates our claim of complementarity and verifies that the dual-task setup yields mutual benefits rather than interference.
>
> ### Response to Weakness 2: "Page-Level" vs. "Column-Level" Generation
> We acknowledge the reviewer's concern regarding the "page-level" description. Our model is indeed trained at the **column level**, a term we have adopted in the revision for precision. However, we argue this is a well-motivated design choice, not a fundamental limitation, for three primary reasons:
>
> 1.  **Artistic Relevance:** The primary aesthetic unit in Chinese calligraphy is the **column**. Artistic flow, inter-character spacing, ligatures, and size variation are all expressed *within* this unit [1, 2]. Our model correctly focuses on this fundamental component.
> 2.  **Trivial Scalability:** As demonstrated in our teaser and appendix figures, our column-level model is trivially extended to generate full, multi-column pages via a simple script. The core generative challenge lies in the column, which our model solves.
> 3.  **Computational Trade-off:** A true, end-to-end page-level model would introduce quadratic complexity and computational costs, making the project intractable and difficult to reproduce. Our approach is a practical and necessary trade-off that captures the artistic essence without sacrificing tractability.
>
> **We have clarified this terminology and rationale in the revised manuscript (Abstract and Introduction).**
>
> **References:**
> [1] Chang J J, Chang T W. Chinese Calligraphy Illustration as Space Form Inspiration[C]//Proceedings of the 8th International Conference on GeoComputation. 2005.
> [2] Wu Y. An analysis of the characters of Chinese calligraphy art based on mathematical elements[J]. Open Journal of Applied Sciences, 2020, 10(2): 25-40.
>
> ### Response to Weakness 3: Style Conditioning and Task Definition
> We thank the reviewer for this point, as it highlights a crucial distinction in our task definition.
>
> * **Task Definition:** This work is **not a general-purpose style transfer model** (i.e., transferring style from an unseen reference image). Instead, it is a **calligraphy foundational model**.
> * **Objective:** The objective is to **accurately and faithfully replicate the distinct styles** of dozens of China's most renowned historical calligraphers (e.g., the *Yan style* of Yan Zhenqing, the *Slender Gold* of Zhao Ji, and the *Cursive Script* (often historically described as "Dead Snake on a Tree") of Huang Tingjian).
> * **Style Control:** This design goal is why style is conditioned on text prompts (i.e., style labels) rather than reference images. The task demands high-fidelity, *named* style replication, not open-ended generalization to arbitrary images.
> * **Value and Validation:** This is a deliberate choice, not a limitation. The model's ability to generate historically accurate calligraphy—a capability **validated by experts from calligraphy associations and numerous enthusiasts**—is its primary contribution. This has significant value for digital cultural heritage preservation, creative industries, and education, where style accuracy and identifiability are paramount.

---

### Author Response · Authors · 2025-11-20
**General Response to Area Chair and All Reviewers**

We sincerely thank the Area Chair and all reviewers for their time and the constructive feedback provided. These insights have been instrumental in refining our manuscript. In addition to our detailed point-by-point responses to each reviewer, we wish to address three overarching themes and highlight the broader value of this work.

### 1. Clarification on Comparative Baselines & The Value of Open Source
A recurring question concerned comparisons with specialized calligraphy generation methods, such as **DP-Font**, **CallifusionV2**, **DeepCalliFont**, and **CalliPaint**.

We wish to emphasize that we did not omit these comparisons by choice. Rather, **none of these works have been made open-source**, rendering a fair and direct comparison impossible. This limitation in the current landscape highlights precisely why our contribution is vital: **we have fully open-sourced our model checkpoint, inference pipeline, and will release our data and training scripts.** We believe this transparency is essential to break the current barrier in the field and enable future researchers to have a reproducible baseline.

### 2. Not Just Style Transfer: A Foundational Model for Digital Heritage
We wish to clarify that our work is not a generic style transfer model; it is a specialized **Column-Level Foundation Model for Ancient Chinese Calligraphy**.

* **Expert Validation:** Our model's ability to authentically reproduce distinct historical styles—such as the dignified *Yan style* of Yan Zhenqing, the exquisite *Slender Gold style* of Zhao Ji, and the oscillating *Cursive Script* of Huang Tingjian—has received widespread recognition from **calligraphy experts and enthusiasts**. We have detailed this qualitative validation in the **revised Appendix H.1 (lines 1073-1090)**.
* **Real-World Impact:** This capability has profound implications for **digital cultural heritage preservation**, **calligraphy education**, and the **creative industries**. To demonstrate this potential, we have updated **Appendix J (page 26-30)** with new examples of creative cultural products designed using our model, showcasing its practical utility beyond academic theory.

### 3. Comprehensive Revisions Based on Feedback
We value the rigor of this review process and have meticulously refined the manuscript to address the wide range of constructive comments received. **Key revisions include, but are not limited to:**

* **Terminology Precision:** Correcting the ambiguity between "page-level" and "column-level" generation throughout the text.
* **Technical Depth:** Adding detailed mathematical formulations and logical explanations for the recognizer’s zero-shot retrieval workflow and the Duplicate RoPE mechanism.
* **Experimental Rigor:** Incorporating new comparative experiments on external benchmarks (CalliBench) and clarifying baseline definitions in ablation studies.
* **Methodological Clarity:** Expanding on design justifications (e.g., MMDiT, Conditional Dropout motivation) and analyzing data-specific artifacts.

Please refer to our individual responses and the revised manuscript for the **complete set of updates**.

### Conclusion: A Shared Vision for the Field
Finally, we would like to reiterate the spirit of this submission. We believe that the Reviewers, the AC, and the authors share a common goal: **to advance the real-world application of AI in traditional art and culture.** Our primary aim is not merely to produce a paper that is textually flawless, but to provide a **tangible, open-source tool** that can genuinely propel the development of the calligraphy industry and the research community. We are confident that the revisions and our commitment to open science make this work a significant step toward that goal.

---

### Meta-Review · Area_Chair_eWiQ · 2026-01-06

**Summary:**

The paper proposes UniCalli, a unified diffusion model for the generation and recognition of Chinese Calligraphy. The model is jointly trained on both generation and recognition tasks. The paper has also proposed a dataset that can benefit future research on the generation of Chinese Calligraphy.

The initial reviews are slightly negative, including concerns about the term "page-level" and the lack of comparisons. In the rebuttal, the authors have addressed most of the major concerns. Given the high-quality results and its technical contribution, I would recommend acceptance of this work. I encourage the authors to incorporate reviewers' suggestions in their next version.

**Reviewer Concerns:**

Most of the major concerns are addressed by the authors. One concern from reviewers is about the ablation of joint training for generation and recognition. It will be nice to have an ablation where a model trained solely on recognition is compared with the full model.

**Reviewer Scores:**

4, 6, 4, 4 --> 6, 6, 4, 6

---

### Decision · Program_Chairs · 2026-01-26

Accept (Poster)